# RAC-LoRA: A Theoretical Optimization Framework for Low-Rank Adaptation

## Abstract

Fine-tuning has become a popular approach to adapting large foundational models to specific tasks. As the size of models and datasets grows, parameter-efficient fine-tuning techniques are increasingly important. One of the most widely used methods is Low-Rank Adaptation (LoRA), with adaptation update expressed as the product of two low-rank matrices. While LoRA was shown to possess strong performance in fine-tuning, it often underperforms when compared to full-parameter fine-tuning (FPFT). Although many variants of LoRA have been extensively studied empirically, their theoretical optimization analysis is heavily under-explored. The starting point of our work is a demonstration that LoRA and its two extensions, Asymmetric LoRA and Chain of LoRA, indeed encounter convergence issues. To address these issues, we propose a general optimization framework that rigorously analyzes the convergence rates of LoRA-based methods. Our approach inherits the empirical benefits of LoRA-style heuristics, but introduces several small but important algorithmic modifications which turn it into a provably convergent method. Our framework serves as a bridge between FPFT and low-rank adaptation. We provide provable guarantees of convergence to the same solution as FPFT, along with the rate of convergence. Additionally, we present a convergence analysis for smooth, non-convex loss functions, covering gradient descent, stochastic gradient descent, and federated learning settings. Our theoretical findings are supported by experimental results.

## 1 Introduction

Many real-world Deep Learning (DL) applications require adapting a large pre-trained model to specific tasks in order to improve its performance (Church et al., 2021). This process, known as fine-tuning, involves adjusting the model from its pre-trained state to better handle the nuances of particular tasks or domains. Fine-tuning is a specialized form of transfer learning, where knowledge gained during pre-training is adapted for new, specific applications (Vrbančič & Podgorelec, 2020).

**Parameter-Efficient Fine-Tuning.** While fine-tuning all model parameters has been effective, modern models with billions of parameters pose significant challenges due to their scale. Full-parameter fine-tuning is often computationally impractical with standard resources. To address this challenge, Parameter-Efficient Fine-Tuning (PEFT) (He et al., 2021) has emerged as a solution, focusing on updating a subset of parameters only (Richtárik & Takáč, 2016), or adding task-specific modules (Xu et al., 2023). PEFT reduces computational costs by modifying fewer parameters or adding external modules, enabling more efficient resource use and lowering storage requirements. This approach significantly reduces both training time and computational demands, making it a practical solution for adapting large models to new tasks (Han et al., 2024).

### 1.1 Low-Rank Adaptation (LoRA)

One of the most popular PEFT methods is Low-Rank Adaptation (LoRA) (Hu et al., 2021). The core idea behind LoRA is that fine-tuning large pre-trained models can be effectively achieved by utilizing lower-dimensional parameter spaces (Li et al., 2018; Aghajanyan et al., 2020). Instead of updating all parameters of a large and potentially dense matrix associated with the weights of a linear layer, LoRA works with the product of two trainable low-rank matrices, which significantly reduces the

number of parameters updated during fine-tuning. These matrices are trained such that their product is added to the pre-trained model weights.

In LoRA (Hu et al., 2021), the weight adaptation is represented as the product of two low-rank matrices (and a scalar multiplier), resulting in the final model

$$W = W^0 + \frac{\alpha}{r} BA,$$

where $W^0 \in \mathbb{R}^{m \times n}$, $B \in \mathbb{R}^{m \times r}$, and $A \in \mathbb{R}^{r \times n}$. Here, $r$ and $\alpha$ respectively denote the LoRA rank and its scaling factor. Typically, since the dimensions of (particularly deep learning) models are enormous, we have rank $r \ll \min\{m, n\}$. This approach saves computational resources and minimizes the risk of overfitting or catastrophic forgetting (Biderman et al., 2024). Hence, LoRA has become a lightweight and efficient technique for adapting large models to various tasks, particularly in resource-constrained environments (Sun et al., 2022). It is important to note that $W^0$ remains fixed and does not receive updates, while $A$ and $B$ are optimized during the training process. The scaling factor $\alpha$ serves as a "step size" for the adaptation, and it is normalized by rank $r$. The matrix $A$ is typically initialized with random Gaussian values, while the matrix $B$ is set to zero, ensuring $\Delta W = 0$ at the start of training. Alternative initialization strategies were explored by Zhu et al. (2024).

### 1.2 CHAIN OF LoRA (COLA)

While LoRA offers significant computational advantages in practice, it remains less effective than full-parameter fine-tuning (FPFT) if efficiency is not a major concern (Biderman et al., 2024). To balance efficiency and performance, Xia et al. (2024) proposed an iterative method called Chain of LoRA (COLA). Essentially, COLA simply means the successive application of several LoRA updates.

Chain of LoRA (COLA) constructs a sequence of LoRA modules through an iterative process of parameter fine-tuning, merging, and extending. The chain length is defined by the number of optimized LoRA modules. COLA's central concept involves applying LoRA adaptations iteratively $T$ times. COLA can be summarized as training a LoRA module, merging the updates with the fixed parameters, reinitializing the LoRA matrices, and repeating the process (Xia et al., 2024). The resulting model can be represented by:

$$W = W^0 + \frac{\alpha}{r} \sum_{t=0}^{T-1} B^t A^t,$$

where $A^t$ and $B^t$ indicate the low-rank matrices in the $t$-th block in the chain, which are typically initialized in the same manner as in standard LoRA. The motivation behind COLA is that standard LoRA may clearly fail to find the optimal adaptation since such an adaptation may not in general be of a low rank. To address this, COLA proposes using a sequence of low-rank matrix decompositions to approximate a middle-to-high-rank update. The hypothesis is that this sequence of updates can provide a better approximation than a single LoRA adaptation and may be easier to optimize compared to learning the optimal adaptation from scratch.

## 2 PROBLEM FORMULATION AND SUMMARY OF CONTRIBUTIONS

### 2.1 PROBLEM FORMULATION

The primary approach for training supervised machine learning models is to formulate the task as an optimization problem where the goal is to minimize a loss function, which measures the discrepancy between the model's predictions and the actual outcomes. In this work, we explore this optimization problem in the specific context of fine-tuning, where a pre-trained model is adapted to a new task or dataset, requiring efficient adjustments to its parameters to achieve better performance on the target task. In particular, we consider the model-agnostic problem formulation

$$\min_{\Delta W \in \mathbb{R}^{m \times n}} f(W^0 + \Delta W), \tag{1}$$

where $W^0 \in \mathbb{R}^{m \times n}$ represents the parameters of a pre-trained model (or of a single linear layer, with the others being fixed), and $\Delta W \in \mathbb{R}^{m \times n}$ denotes the adaptation term. The function $f$ :

Table 1: Sections where we conduct a theoretical convergence analysis of RAC-LoRA for solving Problem (1) when using a specific optimizer for approximately solving the subproblem in Step 4. The results for the RAC-LoRA + GD combination are described in Section 5, while the proofs can be founded in Appendix C. The results and proofs for all other combinations can be found in the indicated appendices.

| Problem | Fine-tuner | Subproblem Optimizer | Non-convex | PL |
|---------|-----------|----------------------|------------|-----|
| (1) | RAC-LoRA | Gradient Descent (GD) | C.1 | C.2 |
| (1) | RAC-LoRA | Random Reshuffling (RR) | D.1 | D.2 |
| (1) | RAC-LoRA | Stochastic Gradient Descent (SGD) | E.1 | E.2 |
| (15) | Fed-RAC-LoRA | Random Reshuffling (RR) | F.1 | F.2 |

$\mathbb{R}^{m \times n} \to \mathbb{R}$ corresponds to the empirical loss over the adaptation dataset, or any other loss function of interest. As the total dimensionality $m \times n$ is typically very large for deep learning models, the adaptation term $\Delta W$ needs to have a specific structure to be feasible in real-world applications.

## 2.2 No Reasonable Theory for Low-Rank Adaptation

We claim that a satisfying theoretical understanding of prevalent fine-tuning methods based on low-rank updates, such as LoRA and COLA, is lacking.

- First, as already noted by Sun et al. (2024), the LoRA re-parameterization of the domain effectively transforms a *smooth* Lipschitz loss into a *non-smooth* Lipschitz loss, which poses *additional* theoretical challenges to those related to proper handling of the low-rank structure of the updates. While this hints at a possible source of issues with providing a good theory for methods based on low-rank adaptation, this observation does not on its own mean that a good theory is impossible to obtain.
- More importantly, the existing theoretical analysis of COLA (Xia et al., 2024) replaces low-rank optimization over matrices $A$ and $B$ with full-rank matrix optimization ($\Delta W$). This makes the theoretical analysis irrelevant at worst and unsatisfactory at best as it completely ignores to model and to explain the key component of LoRA: low-rank updates.
- Third, it is known that LoRA can be highly sensitive to the choice of the hyper-parameters (Khodak et al., 2021; Kuang et al., 2024). A good theory should be able to explain or remove this issue. No such theory exists, to the best of our knowledge.
- Finally, and this is the true starting point of our exploration in this work, we observe that COLA may simply *fail to converge* to the optimal solution. We give a simple example (with $3 \times 3$ matrices) of this divergence behavior in Section 3. Hence, COLA is merely a *heuristic*. Providing a fix is an open problem – the problem we address in this work.

While clearly LoRA and COLA are enormously useful in practice, these methods remain mere *heuristics* since they do not come with solid theoretical backing. This is problematic and raises valid concerns about the robustness and reliability of LoRA-type methods in scenarios *beyond* current datasets, models and practice.

## 2.3 Contributions

To address the aforementioned fundamental issues of LoRA-type heuristics, and to firmly ground the fine-tuning-via-low-rank adaptation line of work in a theoretically sound algorithmic framework, we propose a new generic low-rank adaptation framework for which we coin the name Randomized Asymmetric Chain of LoRA (RAC-LoRA); see Algorithm 1.

- Similarly to COLA (Xia et al., 2024), our method is iterative: we perform a chain of low-rank updates (see Step 2 in Algorithm 1). In each step of the chain, one matrix (e.g., $A$) is chosen randomly from a pre-defined distribution, and the other (e.g., $B$) is trainable (see Step 3 in Algorithm 1). Which of these two update matrices is chosen randomly and which one is trainable is decided a-priori, and hence our method is asymmetric in nature, similarly to AsymmLoRA (Zhu et al., 2024). We propose two options, depending on which matrix is

trainable and which one is chosen randomly: in Option 1, $A$ is trainable, and in Option 2, $B$ is trainable.

- In order to make our framework flexible, we offer a variety of strategies for updating the trainable matrix in each step of the chain. This is possible since in each such step we formulate an auxiliary optimization subproblem in the trainable matrix, and once can thus chose essentially *any optimizer* for approximately solving it (see Step 4 in Algorithm 1). We theoretically analyze several such optimizers within our RAC-LoRA framework, including Gradient Descent (GD) in Appendix C (however, we include and describe the theorems in Section 5.2), Random Reshuffling RR in Appendix D, and Stochastic Gradient Descent (SGD) in Appendix E. See Table 1 for a quick overview. Our analysis applies to the smooth nonconvex regime, in which we prove fast sublinear (i.e., $O(1/T)$) convergence rates to a stationary point, and fast linear (i.e., $O(e^{-T})$) rates to the globally optimal solution under the Polyak-Łojasiewicz (PL) condition.
- The update is applied (see Step 5 in Algorithm 1), and the method moves on to the next step of the chain.

**Experiments.** We apply our method to several machine learning tasks. We start from convex problems with traditional models, such as logistic and linear regression, to provide clear illustrations of our theoretical findings. In addition, we present empirical analyses for multilayer perception (MLP) on MNIST and RoBERTa on the GLUE benchmark tasks (Wang, 2018). See Appendix 6.

**Federated Learning.** Furthermore, we extend our findings from the simple problem (1) to the more challenging distributed/federated problem (15), where we consider solving a distributed optimization problem via our new Fed-RAC-LoRA method (Algorithm 2). These additional results can be found in Section F. For illustrative purposes, we provide an analysis for RR as the optimizer for the subproblem; see also Table 1. Previous research (Sun et al., 2024) has shown that using a single learnable matrix in this context provides several key advantages, particularly in terms of preserving privacy, ensuring the correctness of model aggregation, and maintaining stability when adjusting the scaling factor (Sun et al., 2024). These benefits are crucial in Federated Learning (Konečný et al., 2016), where data is distributed across multiple clients, and privacy constraints must be upheld while performing model updates. Building on this asymmetric approach, we integrate the concept of chained updates to develop Fed-RAC-LoRA, a more robust and scalable distributed method. Our approach maintains the computational efficiency of the original RAC-LoRA while ensuring rigorous convergence properties in the distributed setting, offering a theoretically sound method for large-scale Federated Learning scenarios.

## 3 Shining Some Light on LoRA's Convergence Issues

In contemporary machine learning, loss function minimization is primarily accomplished using gradient-based (first-order) optimization techniques (Ruder, 2016). Most advanced methods build on the vanilla Gradient Descent (GD) in various ways, e.g., by adding support for stochastic approximation, momentum, adaptive stepsizes and more (Shapiro & Wardi, 1996; Gower et al., 2019).

It is therefore meaningful to start our exploration of LoRA-style methods in connection with GD steps. In particular, we analyze the update process of LoRA matrices through a GD step, focusing on the application of the chain rule of differentiation. The gradient with respect to the low-rank matrices $B$ and $A$ consists of two components,

$$\nabla_{B,A} f(W + \frac{\alpha}{r} BA) = \begin{pmatrix} \nabla_A f(W + \frac{\alpha}{r} BA) \\ \nabla_B f(W + \frac{\alpha}{r} BA) \end{pmatrix} = \frac{\alpha}{r} \begin{pmatrix} \nabla B^\top f(W + \frac{\alpha}{r} BA) \\ \nabla f(W + \frac{\alpha}{r} BA) A^\top \end{pmatrix}.$$

and hence the update rules for the matrices $A$ and $B$ are given by

$$A^+ = A - \eta \frac{\alpha}{r} B^\top \nabla f(W + \frac{\alpha}{r} BA), \qquad B^+ = B - \eta \frac{\alpha}{r} \nabla f(W + \frac{\alpha}{r} BA) A^\top,$$

where $\eta > 0$ is a step size, and $A^+$ and $B^+$ are the updated matrices. Since both $A$ and $B$ are trainable, the gradients are multiplied by $B^\top$ and $A^\top$, which adds complexity to the optimization process and complicates the interpretation of its evolution. This interaction between low-rank matrices and gradients creates a non-trivial structure that challenges rigorous analysis and may disrupt Lipschitz

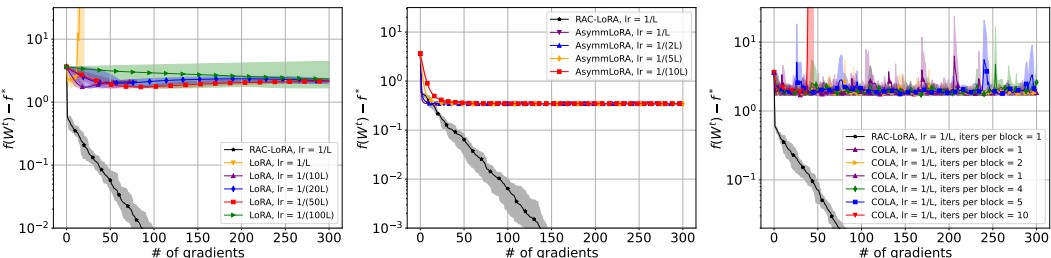

Figure 1: Convergence of LoRA, Asymmetric LoRA (AsymmLoRA), Chain of LoRA (COLA), and our proposed Randomized Asymmetric Chain of LoRA (RAC-LoRA) on the problem in Equation 2.

continuity, raising concerns about convergence guarantees. While LoRA is effective for deep learning adaptation, a deeper understanding of this process is needed to ensure that the optimization scheme is theoretically sound.

**Loss of Lipschitz smoothness.** Lipschitz continuity of the gradient is a commonly invoked assumption in the theoretical analysis of gradient-based optimization methods (Zhou, 2018; Khaled & Richtárik, 2020; Demidovich et al., 2023). This property ensures that the gradient does not change too rapidly, which in turn guarantees a controlled behavior of the optimization process, and plays a key role in establishing convergence rates and in providing stability guarantees for various optimization algorithms (Nesterov, 2004; Sun, 2020). A formal definition follows.

**Assumption 3.1** (Lipschitz Gradient). *Function $f$ is differentiable, and there exists $L > 0$ such that*

$$\|\nabla f(W) - \nabla f(V)\| \leq L\|W - V\|, \qquad \forall W, V \in \mathbb{R}^{m \times n},$$

*where $\|\cdot\|$ denotes the Frobenius matrix norm, the gradient is computed w.r.t. the trace inner product.*

However, the property of Lipschitz smoothness does not necessarily hold when applying LoRA adaptation. Specifically, even if the original function $f(W)$ is Lipschitz smooth, meaning that the gradient of $f(W)$ satisfies the Lipschitz continuity condition (as stated in Assumption 3.1), this smoothness property is generally lost when the function is expressed in the adapted form $f(W^0 + BA)$. In particular, the function $f(W^0 + BA)$ is not Lipschitz smooth with respect to the set of variables $\{B, A\}$ for any constant. This breakdown of smoothness is a significant limitation, as it complicates the theoretical analysis of optimization algorithms when using LoRA. The formal proof of this result is provided in Theorem 2 of the work by Sun et al. (2024), highlighting the challenges in extending standard gradient-based methods to such adaptations.

**Numerical counterexample.** We present a clear and illustrative example demonstrating that the LoRA and COLA methods may not converge to the solution of the optimization problem. To illustrate this, let us consider a quadratic function of the following form:

$$f(x) = x^\top M x + b^\top x, \qquad (2)$$

where $x \in \mathbb{R}^d$ is a vector of parameters, $M \in \mathbb{R}^{d \times d}$ is a positive definite matrix, and $b \in \mathbb{R}^d$ is a vector corresponding to the linear term. In our numerical example, we consider $d = 9$, $M = \mathrm{Diag}(10, 1, 1, 1, 1, 1, 1, 1, 1)$, and $b = (1, 1, 1, 1, 1, 1, 1, 1, 1)^\top$. This function has a Lipschitz gradient (Assumption 3.1) with $L = 10$. We represent the vector $x \in \mathbb{R}^9$ as a matrix $W \in \mathbb{R}^{3 \times 3}$. In the LoRA adaptation, we use a rank $r = 1$ and set $\alpha = r$.

Figure 1 shows experiments on LoRA, AsymmLoRA, our RAC-LoRA, and COLA. In the case of COLA, we varied the step sizes and the number of gradients per block. Our results indicate that, when using the theoretical step size $\frac{1}{L}$, both LoRA and COLA may diverge, while AsymmLoRA converges to a different stationary point. When smaller step sizes are applied to LoRA and COLA, these methods do converge, but to a stationary point that is significantly distant from the optimal solution. In contrast, our RAC-LoRA converges linearly to the optimal solution without such issues. These results provide clear evidence that the choice of LoRA-type updates has a significant impact on both the convergence and the quality of the final solution. The divergence, convergence to suboptimal points,

---

**Algorithm 1** Randomized Asymmetric Chain of LoRA (RAC-LoRA)

---

1: **Parameters:** pre-trained model $W^0 \in \mathbb{R}^{m \times n}$, rank $r \ll \min\{m, n\}$, learning rate $\gamma > 0$, scaling factor $\alpha > 0$, chain length $T$, sketch distribution $\mathcal{D}_S^B$ (Option 1) or $\mathcal{D}_S^A$ (Option 2).

2: **for** $t = 0, 1, \ldots, T - 1$ **do**

3:    Sample a sketch matrix

$$\text{(Option 1)} \quad B_S^t \sim \mathcal{D}_S^B \qquad \text{(Option 2)} \quad A_S^t \sim \mathcal{D}_S^A$$

4:    Using some iterative solver, approximately solve the subproblem

$$\text{(Option 1)} \quad \hat{A}^t \approx \min_A f\left(W^t + \frac{\alpha}{r} B_S^t A\right) \qquad \text{(Option 2)} \quad \hat{B}^t \approx \min_B f\left(W^t + \frac{\alpha}{r} B A_S^t\right)$$

5:    Apply the update

$$\text{(Option 1)} \quad W^{t+1} = W^t + \frac{\alpha}{r} B_S^t \hat{A}^t \qquad \text{(Option 2)} \quad W^{t+1} = W^t + \frac{\alpha}{r} \hat{B}^t A_S^t$$

6: **end for**

---

and sensitivity to step sizes in traditional methods underscore the need for careful selection and design of update mechanisms. Our findings suggest that RAC-LoRA offers a more reliable approach for achieving optimal solutions in the context of LoRA-based adaptations.

## 4 RANDOMIZED ASYMMETRIC CHAIN OF LoRA (RAC-LoRA)

To address the convergence issues in LoRA updates, we propose Randomized Asymmetric Chain of LoRA (RAC-LoRA). This method introduces an asymmetric LoRA mechanism with a chain-based structure to enhance convergence while preserving model flexibility and efficiency. The method is summarized in Algorithm 1.

**Description of the algorithm.** At the start of each iteration (or block), one matrix is randomly initialized and fixed throughout training, while the other remains fully trainable. This strategy prevents optimization within a restricted subspace, reducing the risk of convergence to suboptimal points. There are two configurations: freeze matrix $B$ and train $A$, or freeze $A$ and train $B$. We now formally define the sampling/sketch schemes.

**Definition 4.1** (Left Sketch). *By a "left sketch" (of rank $r$) we refer to the update rule*

$$\Delta W = \frac{\alpha}{r} B_S \hat{A},$$

*where $B_S \sim \mathcal{D}_B$ is sampled from some fixed distribution over matrices of dimensions $n \times r$, and only the matrix $\hat{A}$ is adjustable.*

**Definition 4.2** (Right Sketch). *By a "right sketch" (of rank $r$) we refer to the update rule*

$$\Delta W = \frac{\alpha}{r} \hat{B} A_S,$$

*where $A_S \sim \mathcal{D}_A$ is sampled from some fixed distribution over matrices of dimensions $r \times m$, and only the matrix $\hat{B}$ is adjustable.*

In both sampling schemes, we update the trainable matrix over several epochs. This step effectively corresponds to training a LoRA block within the chain, following the standard LoRA approach. While this procedure mirrors the conventional LoRA method, we can formally characterize it as an approximate optimization problem, allowing for a structured analysis of the training process. These procedures for both matrices can be formally expressed via

$$\text{(Option 1)} \quad \hat{A}^t \approx \min_A f\left(W^t + \frac{\alpha}{r} B_S^t A\right) \qquad \text{(Option 2)} \quad \hat{B}^t \approx \min_B f\left(W^t + \frac{\alpha}{r} B A_S^t\right).$$

Similarly to COLA, $t$ identifies the block in the chain. Next, we incorporate the product of the trained matrix and the sampled matrix into the current model. The merging process involves adding

the product of the two matrices—one sampled and the other trained. This addition is scaled by a factor of $\frac{\alpha}{r}$, ensuring the appropriate weighting of the update within the model:

$$(\text{Option 1}) \quad W^{t+1} = W^t + \frac{\alpha}{r} B_S^t \hat{A}^t \qquad (\text{Option 2}) \quad W^{t+1} = W^t + \frac{\alpha}{r} \hat{B}^t A_S^t.$$

# 5 THEORY

## 5.1 DERIVATION OF THE UPDATE STEP

Without loss of generality, let us focus on the Left Sketch scheme (Definition 4.1). Specifically, for each model in the chain, the update rule is given as follows:

$$W^{t+1} = W^t + \frac{\alpha}{r} B_S^t \hat{A}^t.$$

Next, we apply the Lipschitz gradient condition (Assumption 3.1) to the loss function $f$:

$$f(U) \leq f(V) + \langle \nabla f(V), U - V \rangle + \frac{L}{2} \|U - V\|_F^2, \quad \forall U, V \in \mathbb{R}^{m \times n}$$

Applying this with $U = W^t$, $V = B_S^t \hat{A}^t$ and $\eta \leq \frac{1}{L}$ leads to

$$f(W^{t+1}) \leq f(W^t) + \langle \nabla f(W^t), B_S^t \hat{A}^t \rangle + \frac{L}{2} \|B_S^t \hat{A}^t\|_F^2$$

$$\leq f(W^t) + \langle (B_S^t)^\top \nabla f(W^t), \hat{A}^t \rangle + \frac{1}{2\eta} \langle (B_S^t)^\top B_S^t \hat{A}^t, \hat{A}^t \rangle.$$

Let us minimize the left hand side term in $\hat{A}^t$, when the gradient vanishes: $(B_S^t)^\top \nabla f(W^t) + \frac{1}{\eta}(B_S^t)^\top (B_S^t) \hat{A}^t = 0$. One such solution is given by[1]

$$\hat{A}^t = -\eta \left( (B_S^t)^\top (B_S^t) \right)^\dagger (B_S^t)^\top \nabla f(W^t),$$

and his leads to the following gradient update:

$$W^{t+1} = W^t + \frac{\alpha}{r} B_S^t \hat{A}^t = W^t - \frac{\alpha}{r} \eta B_S^t \left( (B_S^t)^\top (B_S^t) \right)^\dagger (B_S^t)^\top \nabla f(W^t)$$

$$= W^t - \gamma H_B^t \nabla f(W^t), \tag{3}$$

where $H_B^t = B_S^t \left( (B_S^t)^\top (B_S^t) \right)^\dagger (B_S^t)^\top$ is projection matrix and $\frac{\alpha}{r} \eta = \gamma$. Similarly, we can obtain the update for Right Sketch scheme (Definition 4.2):

$$W^{t+1} = W^t - \gamma \nabla f(W^t)(A_S^t)^\top \left( A_S^t (A_S^t)^\top \right)^\dagger A_S^t = W^t - \gamma \nabla f(W^t) H_A^t, \tag{4}$$

where $H_A^t = (A_S^t)^\top \left( A_S^t (A_S^t)^\top \right)^\dagger A_S^t$ is also projection matrix. Notably, the scaling factor $\frac{\alpha}{r}$ is combined with the parameter $\eta$, allowing us to work with the effective step size $\gamma$. This simplifies the learning process by unifying the scaling and learning rate. Using this type of update, we provide convergence results for both standard and stochastic gradient descent methods.

## 5.2 CONVERGENCE RESULTS

To derive the convergence results, a key factor in our analysis is the smallest eigenvalue of the expected value of the projection matrix introduced in Section 5.1. This eigenvalue plays a critical role in shaping the optimization process. As we will show, a well-conditioned projection matrix—with a sufficiently large smallest eigenvalue—ensures more efficient and reliable convergence. Therefore, we make an important assumption that this smallest eigenvalue must remain strictly positive.

**Assumption 5.1.** *Consider a projection matrix $H$ generated by Left Sketch 4.1 or Right Sketch 4.2. Assume that the sampling distributions $\mathcal{D}_S^B$ and $\mathcal{D}_S^A$ are such that the smallest eigenvalue of the expected projection matrix $H$ generated by sampled matrix is positive:*

$$\lambda_{\min}^H = \lambda_{\min} \left[ \mathbb{E}[H] \right] > 0.$$

---

[1]The dagger notation refers to the Moore-Penrose pseudoinverse.

In particular, it is important to observe that the eigenvalues of the projection matrix are either zero or one, with the smallest eigenvalue being zero. However, the smallest eigenvalue of the expected value of the projection matrix can be strictly greater than zero. Additionally, it is essential to establish a lower bound for the loss function.

**Remark.** *Assumption 5.1 is easily satisfied. Let $H$ be the projection matrix as defined below Equation (4) and assume that the $A$ matrices are drawn from an isotropic distribution (the rows of $A$ are isotropic). Then $H$ is the projection onto the rank of $A$, which is a subspace of dimension $r$ distributed isotropically in $\mathbb{R}^n$. The matrix $\mathbb{E}[H]$ is then invariant under rotations, so must be a scalar multiple of the identity. By taking traces, one finds that $\mathbb{E}[H] = \frac{r}{n} I$ so $\lambda_{\min}^H = \frac{r}{n}$.*

**Assumption 5.2.** *Function $f$ is bounded from below by an infimum $f^\star \in \mathbb{R}$.*

We now present the convergence result for RAC-LoRA with Gradient Descent (GD) updates.

**Theorem 5.3.** *Let Assumptions 3.1 and 5.1 hold, and let the stepsize satisfy $0 < \gamma \leq \frac{1}{L}$. Then, the iterates of RAC-LoRA (Algorithm 1) with GD updates (Equation 3 or 4) satisfy*

$$\mathbb{E}\left[\left\|\nabla f(\widetilde{W}^T)\right\|^2\right] \leq \frac{2(f(W^0) - f^\star)}{\lambda_{\min}^H \gamma T},$$

*where the output $\widetilde{W}^T$ is chosen uniformly at random from $W^0, W^1, \ldots, W^{T-1}$.*

We obtain a sub-linear convergence rate, as is expected in general non-convex settings. To achieve a stronger convergence result, we employ an additional assumption: the Polyak-Lojasiewicz (PL) condition. This assumption generalizes strong convexity but applies to certain non-convex functions.

**Assumption 5.4** (PL-condition). *Function $f$ satisfies the Polyak-Łojasiewicz (PL) condition with parameter $\mu > 0$ if*

$$\frac{1}{2}\|\nabla f(W)\|^2 \geq \mu\left(f(W) - f^\star\right)$$

*for all $W \in \mathbb{R}^{m \times n}$, where $f^\star = \inf f$, assumed to be finite.*

Next, we establish a convergence rate for RAC-LoRA in the Polyak-Łojasiewicz setting.

**Theorem 5.5.** *Let Assumptions 3.1, 5.1 and 5.4 hold, and let the stepsize satisfy $0 < \gamma \leq \frac{1}{L}$. Then, for each $T \geq 0$, the iterates of RAC-LoRA (Algorithm 1) with GD updates (Equation 3 or 4) satisfy*

$$\mathbb{E}\left[f(W^T)\right] - f^\star \leq \left(1 - \gamma\mu\lambda_{\min}^H\right)^T \left(f(W^0) - f^\star\right).$$

We achieved a linear convergence rate, which is significantly better than previous results; however, this improvement applies to a more limited class of functions. Importantly, we can recover the classical results of GD by setting $\lambda_{\min}^H = 1$, which corresponds to the full-rank scenario.

The comprehensive analysis of different optimizers and their performance across various settings is provided in the appendix, as summarized in Table 1.

## 6 EXPERIMENTS

In this section, we explore the performance of RAC-LoRA as an optimization algorithm in machine learning applications. In Section 6.1 we validate the theoretical results in convex problems, while in Section 6.2 we evaluate the method applied to neural networks.

### 6.1 CONVEX OPTIMIZATION PROBLEMS

**Linear Regression.** We conducted our analysis in a controlled setting involving linear regression with quadratic regularization applied to synthetic data. Specifically, we utilized 3,000 samples for pre-training the model and 1,000 samples for fine-tuning. In this setup, we have $d = 100$ with weight matrices of size $10 \times 10$, and the regularization term is set to $0.0001$. As illustrated in Figure 2, the method converges for various ranks and the convergence speed is proportional to $\frac{n}{r}$, and when the rank is set to the full rank, we observe convergence identical to that of FPFT. We remark that COLA would suffer from the same divergence behavior as in Figure 1 on this quadratic problem.

**Logistic Regression.** Analogous results for logistic regression are shown in Appendix A.

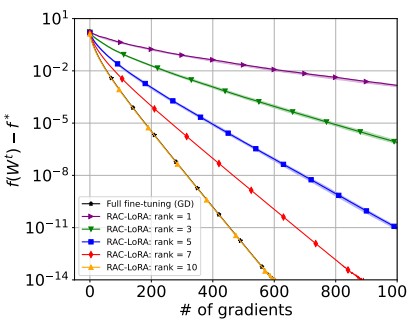 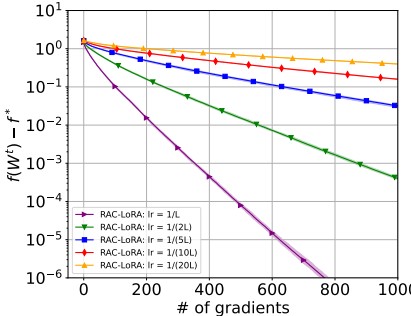

Figure 2: RAC-LoRA convergence with varying ranks and step sizes on a linear regression problem.

## 6.2 NON-CONVEX OPTIMIZATION PROBLEMS

Further experimental results are provided in Appendix B.

### 6.2.1 RESULTS OF ROBERTA ON NLP TASKS

As in prior work (Zhu et al., 2024; Xia et al., 2024), we evaluate low-rank adaptation methods for LLMs using the GLUE dataset (Wang, 2018).

**Methodology.** We fine-tuned the `roberta-base` model (Liu, 2019) on four of the smallest GLUE tasks to study the behavior of low-rank methods in practical scenarios. For the chained methods, we use a range of values for the number of chains and epochs per chain hyperparameters. In each experiment we used rank 2 for the adaptations and trained using the AdamW optimiser (Loshchilov, 2017) with $\beta$ parameters 0.9 and 0.999, $\epsilon = 1 \times 10^{-8}$, a learning rate of $4 \times 10^{-4}$ with linear schedule and a training batch size 8.

**Discussion.** The results are presented in Table 2. We find that RAC-LoRA performs competitively with other low-rank adaptation methods, but does not outperform Asymmetric LoRA despite having greater capacity. We expect RAC-LoRA to outperform Asymmetric LoRA in settings where there is a benefit to the additional capacity, i.e., those where a full parameter fine tune (FPFT) is much better than Asymmetric LoRA. The performance of the FPFT in Table 2 shows that the selected GLUE tasks do not provide such a setting. Here, a single low-rank adaptation is already enough to obtain performance close to that of FPFT. However, this intuition motivates the experiments in Section 6.2.2 where we intentionally restrict capacity of the adaptations to isolate the effect of the chaining procedure.

| Method | # Chains | # Epochs | MRPC | CoLA | RTE | STS-B | Avg |
|---|---|---|---|---|---|---|---|
| FPFT * | 1 | 30, 80, 80, 40 | $90.2_{\pm0.0}$ | $63.6_{\pm0.0}$ | $78.7_{\pm0.0}$ | $91.2_{\pm0.0}$ | 80.9 |
| LoRA * | | | $89.7_{\pm0.7}$ | $63.4_{\pm1.2}$ | $86.6_{\pm0.7}$ | $91.5_{\pm0.2}$ | 82.8 |
| LoRA | 1 | 100 | $87.7_{\pm0.2}$ | $60.8_{\pm0.2}$ | $75.2_{\pm1.5}$ | $90.2_{\pm0.1}$ | 78.5 |
| AsymmLoRA | | | $86.9_{\pm0.3}$ | $58.7_{\pm1.0}$ | $71.0_{\pm3.3}$ | $90.4_{\pm0.0}$ | 76.8 |
| COLA | 10 | 10 | $88.0_{\pm0.8}$ | $59.5_{\pm1.0}$ | $72.1_{\pm0.9}$ | $90.7_{\pm0.2}$ | 77.6 |
| RAC-LoRA | 10 | 10 | $87.0_{\pm0.7}$ | $58.5_{\pm0.1}$ | $72.3_{\pm1.5}$ | $90.3_{\pm0.0}$ | 77.0 |

Table 2: Results with RoBERTa-base for rank 2 on tasks from the GLUE benchmark. *: results taken from the work of Hu et al. (2021). We report Matthews correlation coefficient for COLA, Pearson correlation coefficient for STS-B, and accuracy for the remaining tasks. Results are averaged over 3 seeds and standard deviations are given in the subscript.

### 6.2.2 RESULTS OF MLPs ON MNIST

In this section, we seek to isolate the effect of the chaining procedure on generalisation performance by restricting the capacity of the low-rank adaptations. This ensures that a single adaptation is not sufficient to reach performance comparable with FPFT, allowing us to explore how chaining adaptations can bridge this gap.

**Methodology.** We first pre-train a 3-layer MLP on the first five classes (digits 0-4) and then adapt the network using LoRA-based methods for recognizing the remaining five unseen classes (digits 5-9). The model is evaluated solely on these unseen classes[2]. we used rank 1 for the adaptations and trained using the AdamW optimiser (Loshchilov, 2017) with $\beta$ parameters 0.9 and 0.999, $\epsilon = 1 \times 10^{-8}$, a constant learning rate of $2 \times 10^{-4}$ and a training batch size 128.

Table 3: MLP results on MNIST with rank $r$ and $\alpha$ set to 1. In the case of AsymmLoRA and RAC-LoRA, only the zero-initialized matrix is trained.

| Method | $\mathcal{D}_A$ | $\mathcal{D}_B$ | Acc | Train Params |
|---|---|---|---|---|
| FPFT | - | - | 98.0 | 54,700 |
| LoRA | Gaussian | Zero | 83.8 | 1K |
| COLA | Gaussian | Zero | 92.6 | 1K |
| LoRA | Zero | Gaussian | 87.0 | 1K |
| COLA | Zero | Gaussian | 96.2 | 1K |
| AsymmLoRA | Gaussian | Zero | 62.3 | 133 |
| RAC-LoRA | Gaussian | Zero | 92.0 | 133 |
| AsymmLoRA | Zero | Gaussian | 81.6 | 912 |
| RAC-LoRA | Zero | Gaussian | 96.1 | 912 |

**Discussion.** Table 3 shows results for MNIST with different ranks and initialization. LoRA reaches around 90% of the accuracy of FPFT leaving some margin for improvement when using the chains. COLA constructs a sequence of LoRA modules, delivering significant accuracy improvements over LoRA due to the chaining procedure. The chaining allows COLA to capture richer features (at the cost of training more parameters). However, both LoRA and COLA lack rigorous convergence guarantees. AsymmLoRA has been shown empirically to approximate the performance of LoRA in Sun et al. (2024) — but again no convergence result is provided. Our proposed method (RAC-LoRA) enjoys significant accuracy improvements over AsymmLoRA, again due to the chaining procedure. RAC-LoRA leverages a diverse learning process across different LoRA blocks, which intuitively allows the model to capture a broader range of features. Crucially, RAC-LoRA comes with convergence guarantees (Theorem 5.3 and Theorem 5.5). Finally, we note that each iteration of RAC-LoRA requires training only one matrix per LoRA block, while COLA needs training two matrices. This reduction in trainable parameters may offer advantages in resource-constrained settings, such as Federated Learning, where minimizing communication costs is critical.

## 7 CONCLUSION

In this work, we introduced RAC-LoRA, a framework for parameter-efficient fine-tuning that enables interpolation between low-rank adaptation and full parameter fine-tuning. Motivated by the convergence challenges of LoRA, we propose the iterative algorithm RAC-LoRA and provide convergence guarantees across various settings, including gradient descent, stochastic gradient descent, and random reshuffling. We extended this framework to the federated learning setup, where RAC-LoRA has advantages over competing algorithms in terms of communication efficiency. Finally, we validate our theoretical results empirically in both convex problems, such as linear and logistic regression, and non-convex problems, such as MLPs and LLMs, finding that its chaining procedure is advantageous in settings where standard low-rank adaptation approaches (such as LoRA and AsymmLoRA) fail to capture the richness of a full-parameter fine-tuning.

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

# Appendix

## A RESULTS ON CONVEX OPTIMIZATION PROBLEMS

### A.1 LOGISTIC REGRESSION

We performed our analysis in a controlled environment using logistic regression with quadratic regularization on synthetic data. In this configuration, we set $d = 100$, employed weight matrices of size $10 \times 10$, and used 2,000 samples, with the regularization term fixed at $0.1$. As shown in Figure 3, the method demonstrates convergence across different ranks, and when the rank is set to full rank, we observe convergence that mirrors that of FPFT.

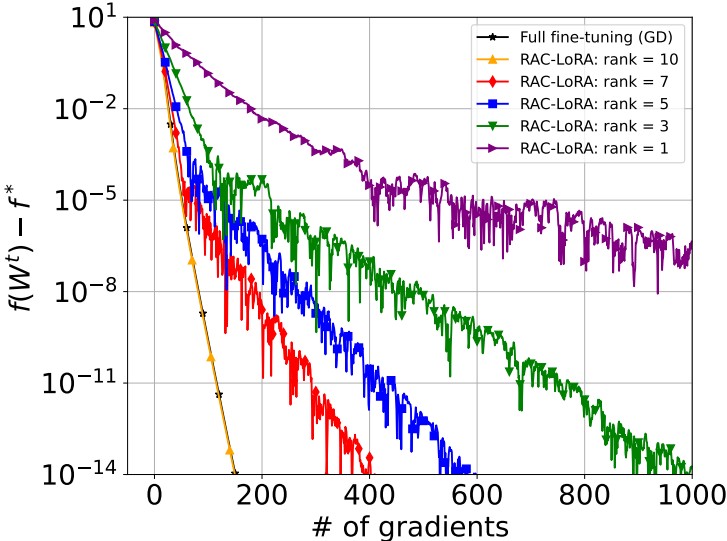

Figure 3: RAC-LoRA convergence with varying ranks and step sizes on a logistic regression problem.

## B RESULTS ON NON-CONVEX OPTIMIZATION PROBLEMS

### B.1 ADDITIONAL RESULTS OF RoBERTa ON NLP TASKS

Table 4 reports additional configurations of the number of epochs per chain and the number of chains on the GLUE benchmark. These results further corroborate the discussion in Section 6.2.

### B.2 ABLATION ON NUMBER OF EPOCHS PER BLOCK IN THE CHAINS

Convergence proof for RAC-LoRA (Corollary D.2.1 and Corollary D.3.1) states that each LoRA module shall be optimized for one epoch only. However, good approximations can also be obtained using more epochs per block and hence fewer blocks (i.e., fewer parameters), as we show in Table 5 for the case of MLP on MNIST.

Similarly, we plot the training loss curves for RoBERTa-base on the RTE dataset in Figure 4. We observe that all setups reach the same value at convergence with similar speed.

| Method | # Chains | # Epochs | MRPC | CoLA | RTE | STS-B | Avg |
|--------|----------|----------|------|------|-----|-------|-----|
| FPFT * | 1 | 30, 80, 80, 40 | $90.2_{\pm0.0}$ | $63.6_{\pm0.0}$ | $78.7_{\pm0.0}$ | $91.2_{\pm0.0}$ | 80.9 |
| LoRA * | | | $89.7_{\pm0.7}$ | $63.4_{\pm1.2}$ | $86.6_{\pm0.7}$ | $91.5_{\pm0.2}$ | 82.8 |
| LoRA | 1 | 20 | $86.8_{\pm0.8}$ | $58.0_{\pm0.4}$ | $71.4_{\pm0.7}$ | $90.3_{\pm0.1}$ | 76.6 |
| AsymmLoRA | | | $85.5_{\pm0.5}$ | $56.5_{\pm1.5}$ | $69.2_{\pm0.2}$ | $89.6_{\pm0.1}$ | 75.2 |
| COLA | 2 | 10 | $87.1_{\pm0.2}$ | $58.4_{\pm1.5}$ | $69.9_{\pm0.9}$ | $90.3_{\pm0.2}$ | 76.4 |
| | 10 | 2 | $84.2_{\pm1.1}$ | $54.2_{\pm0.4}$ | $64.6_{\pm1.3}$ | $89.1_{\pm0.1}$ | 73.0 |
| RAC-LoRA | 2 | 10 | $85.6_{\pm1.7}$ | $55.3_{\pm1.2}$ | $68.6_{\pm1.0}$ | $89.4_{\pm0.2}$ | 74.7 |
| | 10 | 2 | $85.4_{\pm0.4}$ | $55.1_{\pm1.2}$ | $65.5_{\pm0.9}$ | $89.3_{\pm0.1}$ | 73.8 |
| LoRA | 1 | 50 | $88.2_{\pm0.3}$ | $60.1_{\pm0.4}$ | $74.4_{\pm0.9}$ | $90.6_{\pm0.1}$ | 78.3 |
| AsymmLoRA | | | $86.4_{\pm1.0}$ | $57.4_{\pm0.3}$ | $69.9_{\pm1.8}$ | $90.3_{\pm0.1}$ | 76.0 |
| COLA | 5 | 10 | $87.8_{\pm1.1}$ | $59.3_{\pm2.1}$ | $71.2_{\pm1.2}$ | $90.6_{\pm0.2}$ | 77.2 |
| | 10 | 5 | $87.7_{\pm0.5}$ | $58.1_{\pm1.2}$ | $70.9_{\pm0.5}$ | $90.2_{\pm0.2}$ | 76.7 |
| RAC-LoRA | 5 | 10 | $87.2_{\pm0.6}$ | $57.6_{\pm0.5}$ | $70.6_{\pm0.7}$ | $90.2_{\pm0.1}$ | 76.4 |
| | 10 | 5 | $87.5_{\pm0.4}$ | $57.8_{\pm1.0}$ | $70.3_{\pm1.2}$ | $90.2_{\pm0.2}$ | 76.5 |
| LoRA | 1 | 100 | $87.7_{\pm0.2}$ | $60.8_{\pm0.2}$ | $75.2_{\pm1.5}$ | $90.2_{\pm0.1}$ | 78.5 |
| AsymmLoRA | | | $86.9_{\pm0.3}$ | $58.7_{\pm1.0}$ | $71.0_{\pm3.3}$ | $90.4_{\pm0.0}$ | 76.8 |
| COLA | 10 | 10 | $88.0_{\pm0.8}$ | $59.5_{\pm1.0}$ | $72.1_{\pm0.9}$ | $90.7_{\pm0.2}$ | 77.6 |
| RAC-LoRA | 10 | 10 | $87.0_{\pm0.7}$ | $58.5_{\pm0.1}$ | $72.3_{\pm1.5}$ | $90.3_{\pm0.0}$ | 77.0 |

Table 4: Performance of the methods using RoBERTa-base for rank 2. The experiments are based on 4 tasks from the GLUE benchmark. * denotes the results reported in Hu et al. (2021). We report Matthews correlation coefficient for the CoLA dataset, Pearson correlation coefficient for STS-B, and accuracy for the remaining tasks, with the standard deviations given in the subscript. The results are obtained using 3 random seeds.

| | **Number of epochs per block** | | | | | |
|--------|------|------|------|------|------|------|
| | **1** | **2** | **3** | **4** | **5** | **10** |
| COLA | 96.2 | 95.8 | 95.9 | 95.1 | 95.4 | 94.5 |
| RAC-LORA | 96.1 | 95.6 | 95.6 | 94.9 | 94.7 | 93.9 |

Table 5: Accuracy at varying epochs for each block in the chained methods (COLA and RAC-LORA). The setup is the same as in Table 3, with a zero-initialized $A$ matrix and a Gaussian-initialized $B$ matrix. To ensure a fair comparison, the product of the number of epochs per block and the number of blocks is kept constant at 50. The number of trainable parameters for COLA and RAC-LORA are 1K and 912, respectively.

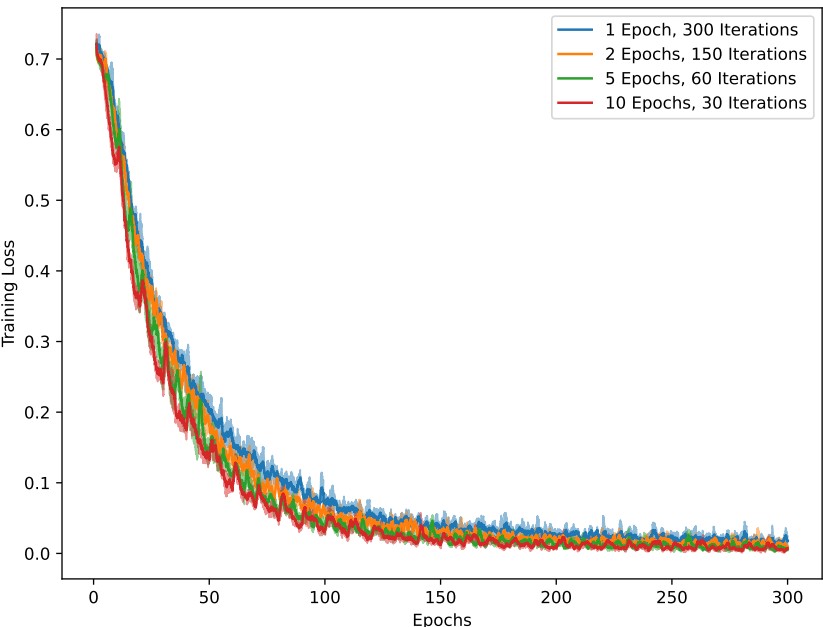

Figure 4: RAC-LORA training loss curves at a fixed computational budget for varying epochs for each block in the chain. RoBERTa-base with rank 2.

# C    ANALYSIS OF RAC-LoRA WITH GRADIENT DESCENT

## C.1    PROOF OF THEOREM 5.3

**Theorem.** *Suppose that Assumption 3.1 and Assumption 5.1 hold. Suppose that a stepsize $\gamma > 0$ is chosen such that $\gamma \leq \frac{1}{L}$. We choose the output of the method $\widetilde{W}^T$ uniformly at random from $W^0, W^1, \ldots, W^{T-1}$ Then, the iterate $\widetilde{W}^T$ of* RAC-LoRA *method (Algorithm 1) with* GD *updates (Equation 3 or Equation 4) satisfy*

$$\mathbb{E}\left[\left\|\nabla f(\widetilde{W}^T)\right\|^2\right] \leq \frac{2f(W^0) - f^\star}{\lambda_{\min}^H \gamma T}.$$

The proof is provided for Left Sketch (Definition 4.1). The result for Right Sketch (Definition 4.2) can be derived by following the same steps.

*Proof.* We begin by examining the implications of Assumption 3.1. The relationships between various conditions associated with Assumption 3.1 are discussed in detail in Nesterov (2004).

$$f(W^{t+1}) \leq f(W^t) + \left\langle \nabla f(W^t), W^{t+1} - W^t \right\rangle + \frac{L}{2} \left\| W^{t+1} - W^t \right\|^2$$

Using the update rule $W^{t+1} = W^t - \gamma H_B^t \nabla f(W^t)$ we get

$$f(W^{t+1}) \leq f(W^t) + \left\langle \nabla f(W^t), -\gamma H_B^t \nabla f(W^t) \right\rangle + \frac{L}{2} \left\| -\gamma H_B^t \nabla f(W^t) \right\|^2$$

$$\leq f(W^t) - \gamma \left\langle \nabla f(W^t), H_B^t \nabla f(W^t) \right\rangle + \frac{L}{2}\gamma^2 \left\| H_B^t \nabla f(W^t) \right\|^2$$

$$\leq f(W^t) - \gamma \left\langle \nabla f(W^t), H_B^t \nabla f(W^t) \right\rangle + \frac{L}{2}\gamma^2 \left\langle H_B^t \nabla f(W^t), H_B^t \nabla f(W^t) \right\rangle$$

$$\leq f(W^t) - \gamma \left\langle \nabla f(W^t), H_B^t \nabla f(W^t) \right\rangle + \frac{L}{2}\gamma^2 \left\langle \nabla f(W^t), (H_B^t)^\top H_B^t \nabla f(W^t) \right\rangle.$$

Since matrix $H_B^t$ is projection matrix, we have $(H_B^t)^\top H_B^t = (H_B^t)^2 = H_B^t$:

$$f(W^{t+1}) \leq f(W^t) - \gamma \left\langle \nabla f(W^t), H_B^t \nabla f(W^t) \right\rangle + \frac{L}{2}\gamma^2 \left\langle \nabla f(W^t), H_B^t \nabla f(W^t) \right\rangle.$$

Using the fact that $\gamma \leq \frac{1}{L}$ we have

$$f(W^{t+1}) \leq f(W^t) - \frac{\gamma}{2} \left\langle \nabla f(W^t), H_B^t \nabla f(W^t) \right\rangle.$$

Taking expectation we get

$$\mathbb{E}\left[ f(W^{t+1}) \mid W^t \right] \leq \mathbb{E}\left[ f(W^t) - \frac{\gamma}{2} \left\langle \nabla f(W^t), H_B^t \nabla f(W^t) \right\rangle \mid W^t \right]$$

$$\leq f(W^t) - \frac{\gamma}{2} \left\langle \nabla f(W^t), \mathbb{E}\left[ H_B^t \right] \nabla f(W^t) \right\rangle$$

Using an Assumption 5.1 we have

$$\mathbb{E}\left[ f(W^{t+1}) \mid W^t \right] \leq \mathbb{E}\left[ f(W^t) - \frac{\gamma}{2} \left\langle \nabla f(W^t), H_B^t \nabla f(W^t) \right\rangle \mid W^t \right]$$

$$\leq f(W^t) - \frac{\gamma}{2} \lambda_{\min}^{H_B} \left\| \nabla f(W^t) \right\|^2.$$

Subtracting $f^\star$ from both sides we get

$$\mathbb{E}\left[ f(W^{t+1}) \mid W^t \right] - f^\star \leq f(W^t) - f^\star - \frac{\gamma}{2} \lambda_{\min}^{H_B} \left\| \nabla f(W^t) \right\|^2. \tag{5}$$

Now we can rewrite as

$$\frac{\gamma}{2} \lambda_{\min}^{H_B} \left\| \nabla f(W^t) \right\|^2 \leq \left( f(W^t) - f^\star \right) - \left( \mathbb{E}\left[ f(W^{t+1}) \mid W^t \right] - f^\star \right)$$

Taking expectation and using tower property we obtain

$$\frac{\gamma}{2} \lambda_{\min}^{H_B} \mathbb{E}\left[\left\|\nabla f(W^t)\right\|^2\right] \leq e^t - e^{t+1},$$

where $e^t = \mathbb{E}\left[f(W^t)\right] - f^\star$. Now we can sum these inequalities together and get

$$\sum_{t=0}^{T-1} \frac{\gamma}{2} \lambda_{\min}^{H_B} \mathbb{E}\left[\left\|\nabla f(W^t)\right\|^2\right] \leq \sum_{t=0}^{T-1} \left(e^t - e^{t+1}\right),$$

Using telescoping property of $e^t - e^{t+1}$ we get

$$\sum_{t=0}^{T-1} \frac{\gamma}{2} \lambda_{\min}^{H_B} \mathbb{E}\left[\left\|\nabla f(W^t)\right\|^2\right] \leq e^0 - e^T.$$

Once we divide by $T$ we obtain

$$\frac{1}{T} \sum_{t=0}^{T-1} \frac{\gamma}{2} \lambda_{\min}^{H_B} \mathbb{E}\left[\left\|\nabla f(W^t)\right\|^2\right] \leq \frac{e^0 - e^T}{T}$$

$$\leq \frac{e^0}{T}.$$

Finally, we get

$$\frac{1}{T} \sum_{t=0}^{T-1} \mathbb{E}\left[\left\|\nabla f(W^t)\right\|^2\right] \leq \frac{2(f(W^0) - f^\star)}{\lambda_{\min}^{H_B} \gamma T}.$$

Applying argument from Danilova et al. (2022) we obtain the result for uniformly chosen point.

$\square$

## C.2 Proof of Theorem 5.5

**Theorem.** *Suppose that Assumption 3.1, Assumption 5.4 and Assumption 5.1 hold. Suppose that a stepsize $\gamma \geq 0$ is chosen such that $\gamma \leq \frac{1}{L}$. Then, the iterates of* RAC-LoRA *method (Algorithm 1) with* GD *updates (Equation 3 or Equation 4) satisfy*

$$\mathbb{E}\left[f(W^T)\right] - f^\star \leq \left(1 - \gamma\mu\lambda_{\min}^H\right)^T \left(f(W^0) - f^\star\right).$$

The proof is provided for Left Sketch (Definition 4.1). The result for Right Sketch (Definition 4.2) can be derived by following the same steps.

*Proof.* We start from the inequality 5:

$$\mathbb{E}\left[f(W^{t+1}) \mid W^t\right] - f^\star \leq f(W^t) - f^\star - \frac{\gamma}{2}\lambda_{\min}^{H_B}\left\|\nabla f(W^t)\right\|^2.$$

Using PL condition $\left\|\nabla f(W^t)\right\|^2 \geq 2\mu\left(f(W^t) - f^\star\right)$ we have

$$\mathbb{E}\left[f(W^{t+1}) \mid W^t\right] - f^\star \leq f(W^t) - f^\star - \gamma\mu\lambda_{\min}^{H_B}\left(f(W^t) - f^\star\right)$$

$$\leq \left(1 - \gamma\mu\lambda_{\min}^{H_B}\right)\left(f(W^t) - f^\star\right).$$

Once we unroll the recursion we get

$$\mathbb{E}\left[f(W^T)\right] - f^\star \leq \left(1 - \gamma\mu\lambda_{\min}^{H_B}\right)^T \left(f(W^0) - f^\star\right).$$

In order to obtain $\varepsilon$ solution we need to take

$$T \geq \mathcal{O}\left(\frac{L}{\mu} \frac{1}{\lambda_{\min}^{H_B}} \log \frac{1}{\varepsilon}\right).$$

$\square$

# D ANALYSIS OF RAC-LoRA WITH RANDOM RESHUFFLING

The previous results were obtained using full gradients. However, this approach is impractical in deep learning settings, where calculating full gradients is often infeasible. To analyze stochastic methods, we assume a finite sum structure for the loss function:

$$\min_{\Delta W \in \mathbb{R}^{m \times n}} \left[ f(W^0 + \Delta W) = \frac{1}{N} \sum_{i=1}^{N} f_i(W^0 + \Delta W) \right], \tag{6}$$

where each function $f_i$ represents the individual loss function for one sample and $N$ is total number of datapoints. Next, we analyze a practical variant of stochastic gradient descent (SGD) known as Random Reshuffling (RR), which involves sampling without replacement. In this method, the dataset is shuffled according to a permutation, ensuring that each training sample is used exactly once during each epoch.

RR is a variant of SGD in which each data point is used exactly once per epoch, also known as SGD with sampling without replacement. Many efforts have been made to explain why gradient methods with reshuffling perform so well in practice, across different types of problems. The convergence rates for incremental gradient methods with random reshuffling in convex optimization were first explored by Nedić & Bertsekas (2001) and later by Bertsekas (2011). In recent years, a lot of focus has shifted toward strongly convex problems, with studies showing that RR can outperform SGD. For example, Recht & Ré (2012) were among the first to analyze this for quadratic least squares problems.

Researchers have also managed to improve results and remove some of the earlier assumptions, such as second-order smoothness, as seen in works by Jain et al. (2019), Safran & Shamir (2021) and Mishchenko et al. (2020). These studies introduced a new way to account for the random permutation's variance, making it easier to analyze both convex and strongly convex cases. There have even been extensions into non-convex settings, with results under the PL condition (Ahn et al., 2020; Nguyen et al., 2021) and general non-convex smooth cases (Lu et al., 2022; Mishchenko et al., 2020; Malinovsky et al., 2023c). More recently, tighter lower bounds for strongly convex and PL functions have been developed (Cha et al., 2023).

In recent years, there's also been growing interest in applying these reshuffling techniques to distributed and federated learning, which is crucial for training large-scale, decentralized models (Yun et al., 2021; Malinovsky et al., 2023b; Sadiev et al., 2022b; Mishchenko et al., 2022a; Cho et al., 2023; Malinovsky & Richtárik, 2022; Malinovsky et al., 2023a; Horváth et al., 2022)

To analyze stochastic methods, we need to make assumptions about the variance. The standard assumption is that the variance is bounded:

**Assumption D.1.** *There exist nonnegative constants $\sigma \geq 0$ such that for any $W^t \in \mathbb{R}^{m \times n}$ we have,*

$$\frac{1}{N} \sum_{i=1}^{n} \left\| \nabla f_i \left( W^t \right) - \nabla f \left( W^t \right) \right\|^2 \leq \sigma^2.$$

The proof is provided for Left Sketch (Definition 4.1). The result for Right Sketch (Definition 4.2) can be derived by following the same steps.

We consider a method belonging to the class of data permutation methods which is the RR algorithm. In each epoch $t$ of RR, we sample indices $\pi_0, \pi_1, \ldots, \pi_{N-1}$ without replacement from $\{1, 2, \ldots, N\}$, i.e., $\{\pi_0, \pi_1, \ldots, \pi_{N-1}\}$ is a random permutation of the set $\{1, 2 \ldots N\}$ and proceed with $N$ iterates of the form:

$$W_{i+1}^t = W_i^t - \gamma H_B^t \nabla f(W_i^t).$$

We then set $W^{t+1} = W_N^t$, and repeat the process for a total of $T$ LoRA blocks. We can derive the effective step:

$$W^{t+1} = W^t - \gamma H_B^t \sum_{i=0}^{N-1} \nabla f(W_i^t) = W^t - \gamma H_B^t N \hat{g}^t, \tag{7}$$

where $\hat{g}^t = \frac{1}{N} \sum_{i=0}^{N-1} \nabla f(W_i^t)$.

## D.1 ANALYSIS OF GENERAL NON-CONVEX SETTING

**Theorem D.2.** *Suppose that Assumption 3.1 and Assumption 5.1 hold. Suppose that a stepsize $\gamma > 0$ is chosen such that $\gamma \leq \frac{1}{2LN}$. We choose the output of the method $\widetilde{W}^T$ uniformly at random from $W^0, W^1, \ldots, W^{T-1}$ Then, the iterate $\widetilde{W}^T$ of RAC-LoRA method (Algorithm 1) with RR updates (Equation 7) satisfy*

$$\mathbb{E}\left[\left\|\nabla f(\widetilde{W}^T)\right\|^2\right] \leq \frac{2}{\gamma N T} \frac{f(W^0) - f^\star}{\left(1 - \lambda_{\max}\left[\mathbb{E}\left[I - H^t\right]\right] - \frac{1}{4}\lambda_{\max}^H\right)}$$
$$+ \frac{L^2 \gamma^2 \lambda_{\max}^H N \sigma^2}{\left(1 - \lambda_{\max}\left[\mathbb{E}\left[I - H^t\right]\right] - \frac{1}{4}\lambda_{\max}^H\right)}.$$

*Proof.* In this context, and in subsequent discussions, the notation $\|\cdot\|$ refers to the Frobenius norm, while $\langle\cdot\rangle$ denotes the inner product associated with the Frobenius norm.

Now we can apply the $L$-smoothness:

$$f(W^{t+1}) \leq f(W^t) + \left\langle \nabla f(W^t), W^{t+1} - W^t \right\rangle + \frac{L}{2}\left\|W^{t+1} - W^t\right\|^2$$

$$= f(W^t) + \left\langle \nabla f(W^t), -\gamma H_B^t N \hat{g}^t \right\rangle + \frac{L}{2}\left\|\gamma H_B^t N \hat{g}^t\right\|^2$$

$$= f(W^t) - \gamma N \left\langle \nabla f(W^t), H_B^t \hat{g}^t \right\rangle + \frac{L}{2}\gamma^2 N^2 \left\|H_B^t \hat{g}^t\right\|^2$$

$$= f(W^t) - \frac{\gamma N}{2}\left(\left\|\nabla f(W^t)\right\|^2 + \left\|H_B^t \hat{g}^t\right\|^2 - \left\|\nabla f(W^t) - H_B^t \hat{g}^t\right\|^2\right) + \frac{L}{2}\gamma^2 N^2 \left\|H_B^t \hat{g}^t\right\|^2$$

$$= f(W^t) - \frac{\gamma N}{2}\left(\left\|\nabla f(W^t)\right\|^2 + \left\|H_B^t \hat{g}^t\right\|^2 - \left\|\nabla f(W^t) - H_B^t \hat{g}^t\right\|^2\right) + \frac{L}{2}\gamma^2 N^2 \left\|H_B^t \hat{g}^t\right\|^2$$

$$= f(W^t) - \frac{\gamma N}{2}\left\|\nabla f(W^t)\right\|^2 - \frac{\gamma N}{2}\left\|H_B^t \hat{g}^t\right\|^2 (1 - \gamma L N) + \frac{\gamma N}{2}\left\|\nabla f(W^t) - H_B^t \hat{g}^t\right\|^2.$$

Using $\gamma \leq \frac{1}{LN}$ we get

$$f(W^{t+1}) \leq f(W^t) - \frac{\gamma N}{2}\left\|\nabla f(W^t)\right\|^2 + \frac{\gamma N}{2}\left\|\nabla f(W^t) - H_B^t \hat{g}^t\right\|^2.$$

Let us take expectation and subtract $f^\star$:

$$\mathbb{E}\left[f(W^{t+1}) \mid W^t\right] - f^\star \leq f(W^t) - f^\star - \frac{\gamma N}{2}\left\|\nabla f(W^t)\right\|^2 + \frac{\gamma N}{2}\mathbb{E}\left[\left\|\nabla f(W^t) - H_B^t \hat{g}^t\right\|^2 \mid W^t\right].$$

Let us consider the last term:

$$\mathbb{E}\left[\left\|\nabla f(W^t) - H_B^t \hat{g}^t\right\|^2 \mid W^t\right]$$

$$= \mathbb{E}\left[\left\|\frac{1}{N}\sum_{i=0}^{N-1}\nabla f_{\pi_i}(W^t) - H_B^t \frac{1}{N}\sum_{i=0}^{N-1}\nabla f_{\pi_i}(W_i^t)\right\|^2 \mid W^t\right]$$

$$= \mathbb{E}\left[\left\|\frac{1}{N}\sum_{i=0}^{N-1}\nabla f_{\pi_i}(W^t) + H_B^t \frac{1}{N}\sum_{i=0}^{N-1}\nabla f_{\pi_i}(W^t) - H_B^t \frac{1}{N}\sum_{i=0}^{N-1}\nabla f_{\pi_i}(W^t) - H_B^t \frac{1}{N}\sum_{i=0}^{N-1}\nabla f_{\pi_i}(W_i^t)\right\|^2 \mid W^t\right]$$

Since $I - H_B^t$ and $H_B^t$ are projection matrices generating perpendicular subspaces we have

$$\mathbb{E}\left[\left\|\nabla f(W^t) - H_B^t \hat{g}^t\right\|^2 \mid W^t\right]$$

$$= \mathbb{E}\left[\left\|\left(I - H_B^t\right)\nabla f(W^t)\right\|^2 + \left\|H_B^t \frac{1}{n}\sum_{i=0}^{n-1}\left(\nabla f_{\pi_i}(W^t) - f_{\pi_i}(W_i^t)\right)\right\|^2 \mid W^t\right]$$

$$= \mathbb{E}\left[\left\langle\left(I - H_B^t\right)\nabla f(W^t), \left(I - H_B^t\right)\nabla f(W^t)\right\rangle + \left\|H_B^t \frac{1}{N}\sum_{i=0}^{N-1}\left(\nabla f_{\pi_i}(W^t) - f_{\pi_i}(W_i^t)\right)\right\|^2 \mid W^t\right].$$

Using the property that $H_B^t$ and $I - H_B^t$ are projection matrices we obtain

$$
\mathbb{E}\left[\left\|\nabla f(W^t) - H^t \hat{g}^t\right\|^2 \mid W^t\right]
$$

$$
\leq \lambda_{\max}\left[\mathbb{E}\left[I - H^t\right]\right]\left\|\nabla f(W^t)\right\|^2 + \mathbb{E}\left[\lambda_{\max}[H^t]L^2\frac{1}{N}\sum_{i=0}^{N-1}\left\|W^t - W_i^t\right\|^2\right].
$$

Since $\lambda_{\max}[H^t] = 1$ for projections matrix we get

$$
\mathbb{E}\left[\left\|\nabla f(W^t) - H_B^t \hat{g}^t\right\|^2 \mid W^t\right] \leq \lambda_{\max}\left[\mathbb{E}\left[I - H_B^t\right]\right]\left\|\nabla f(W^t)\right\|^2 + L^2\frac{1}{N}\sum_{i=0}^{N-1}\mathbb{E}\left[\left\|W^t - W_i^t\right\|^2 \mid W^t\right].
$$

Now let us consider the last term:

$$
\mathbb{E}\left[\left\|W^t - W_k^t\right\|^2\right] = \gamma^2 \mathbb{E}\left[\left\|\sum_{i=0}^{k-1} H_B^t \nabla f_{\pi_i}(W_i^t)\right\|^2 \mid W^t\right]
$$

$$
= \gamma^2 \mathbb{E}\left[\left\|\sum_{i=0}^{k-1} H_B^t\left(\nabla f_{\pi_i}(W_i^t) - \nabla f_{\pi_i}(W^t)\right) + \sum_{i=0}^{k-1} H_B^t \nabla f_{\pi_i}(W^t)\right\|^2 \mid W^t\right]
$$

$$
\leq 2\gamma^2 k \mathbb{E}\left[\sum_{i=0}^{k-1}\left(\left\|H_B^t\left(\nabla f_{\pi_i}(W_i^t) - \nabla f_{\pi_i}(W^t)\right)\right\|^2 + 2\gamma^2 k^2 \left\|H_B^t \nabla f_{\pi_i}(W^t)\right\|^2\right) \mid W^t\right]
$$

$$
\leq 2\gamma^2 k \mathbb{E}\left[\sum_{i=0}^{k-1}\left(\lambda_{\max}\left[H^t\right]\left\|W_i^t - W^t\right\|^2 + 2\gamma^2 k^2 \left\|H_B^t \nabla f_{\pi_i}(W^t)\right\|^2\right) \mid W^t\right]
$$

$$
\leq 2\gamma^2 k \mathbb{E}\left[\sum_{i=0}^{k-1}\left(\left\|W_i^t - W^t\right\|^2 + 2\gamma^2 k^2 \lambda_{\max}\left[\mathbb{E}\left[H_B^t\right]\right]\left\|\nabla f_{\pi_i}(W^t)\right\|^2\right) \mid W^t\right].
$$

Now, we are ready to sum the inequalities. By using $\lambda_{\max}\left[\mathbb{E}\left[H^t\right]\right] = \lambda_{\max}^{H_B}$ and applying Lemma 1 from Mishchenko et al. (2020) with Assumption D.1, we obtain:

$$
\sum_{i=0}^{n-1} \mathbb{E}\left[\left\|W^t - W_i^t\right\|^2\right] \leq \mathbb{E}\left[\sum_{i=0}^{N-1}\left(2\gamma^2 k \sum_{i=0}^{k-1}\left\|W_i^t - W^t\right\|^2 + 2\gamma^2 k^2 \lambda_{\max}^{H_B}\left\|\nabla f_{\pi_i}(W^t)\right\|^2\right) \mid W^t\right]
$$

$$
\leq \gamma^2 L^2 N(N-1)\sum_{i=0}^{N-1}\mathbb{E}\left[\left\|W^t - W_k^t\right\|^2\right]
$$

$$
+ \frac{1}{3}\gamma^2(N-1)N(2N-1)\lambda_{\max}^{H_B}\left\|\nabla f(W^t)\right\|^2 + \frac{1}{3}\lambda_{\max}^{H_B}\gamma^2 N(N+1)\sigma^2.
$$

Using $\gamma \leq \frac{1}{2LN}$ we get

$$
\sum_{i=0}^{n-1} \mathbb{E}\left[\left\|W^t - W_i^t\right\|^2\right] \leq \frac{4}{3}\left(1 - \gamma^2 L^2 N(N-1)\right)\sum_{i=0}^{N-1}\mathbb{E}\left[\left\|W^t - W_i^t\right\|^2\right]
$$

$$
\leq \frac{4}{3}\left(\frac{1}{3}\gamma^2(N-1)N(2N-1)\lambda_{\max}^{H_B}\left\|\nabla f(W^t)\right\|^2 + \frac{1}{3}\lambda_{\max}^{H_B}\gamma^2 N(N+1)\sigma^2\right)
$$

$$
\leq \gamma^2 n^3 \lambda_{\max}^{H_B}\left\|\nabla f(W^t)\right\|^2 + \gamma^2 \lambda_{\max}^{H_B} N^2 \sigma^2
$$

Plugging to the previous bound we obtain:

$$
\mathbb{E}\left[\left\|\nabla f(W^t) - H^t \hat{g}^t\right\|^2 \mid W^t\right] \leq \lambda_{\max}\left[\mathbb{E}\left[I - H_B^t\right]\right]\left\|\nabla f(W^t)\right\|^2 + L^2\gamma^2 N^2 \lambda_{\max}^{H_B}\left\|\nabla f(W^t)\right\|^2
$$

$$
+ L^2\gamma^2 \lambda_{\max}^{H_B} N\sigma^2.
$$

Now we have the following

$$\mathbb{E}\left[f(W^{t+1}) \mid W^t\right] - f^\star \leq f(W^t) - f^\star - \frac{\gamma N}{2}\left\|\nabla f(W^t)\right\|^2$$

$$+ \frac{\gamma N}{2}\left(\lambda_{\max}\left[\mathbb{E}\left[I - H_B^t\right]\right]\left\|\nabla f(W^t)\right\|^2 + L^2\gamma^2 N^2 \lambda_{\max}^{H_B}\left\|\nabla f(W^t)\right\|^2\right)$$

$$+ \frac{\gamma N}{2}L^2\gamma^2\lambda_{\max}^{H_B}N\sigma^2.$$

Using $\gamma \leq \frac{1}{2LN}$ we get

$$\mathbb{E}\left[f(W^{t+1}) \mid W^t\right] - f^\star \leq f(W^t) - f^\star - \frac{\gamma N}{2}\left\|\nabla f(W^t)\right\|^2\left(1 - \lambda_{\max}\left[\mathbb{E}\left[I - H_B^t\right]\right] - \frac{1}{4}\lambda_{\max}^{H_B}\right) \tag{8}$$

$$+ \frac{\gamma N}{2}L^2\gamma^2\lambda_{\max}^{H_B}N\sigma^2. \tag{9}$$

After rearranging the terms, we have

$$\frac{\gamma N}{2}\left\|\nabla f(W^t)\right\|^2\left(1 - \lambda_{\max}\left[\mathbb{E}\left[I - H_B^t\right]\right] - \frac{1}{4}\lambda_{\max}^{H_B}\right) \leq \left(f(W^t) - f^\star\right) - \left(\mathbb{E}\left[f(W^{t+1}) \mid W^t\right] - f^\star\right)$$

$$+ \frac{\gamma N}{2}L^2\gamma^2\lambda_{\max}^{H_B}N\sigma^2.$$

Next, we have

$$\left\|\nabla f(W^t)\right\|^2 \leq \frac{2}{\gamma N}\frac{1}{\left(1 - \lambda_{\max}\left[\mathbb{E}\left[I - H_B^t\right]\right] - \frac{1}{4}\lambda_{\max}^{H_B}\right)}\left(\left(f(W^t) - f^\star\right) - \left(\mathbb{E}\left[f(W^{t+1}) \mid W^t\right] - f^\star\right)\right)$$

$$+ \frac{2}{\gamma N}\frac{1}{\left(1 - \lambda_{\max}\left[\mathbb{E}\left[I - H_B^t\right]\right] - \frac{1}{4}\lambda_{\max}^{H_B}\right)}\frac{\gamma N}{2}L^2\gamma^2\lambda_{\max}^{H_B}N\sigma^2.$$

Using telescoping property and taking expectation we get

$$\frac{1}{T}\sum_{t=0}^{T-1}\left\|\nabla f(W^t)\right\|^2 \leq \frac{2}{\gamma NT}\frac{f(W^0) - f^\star}{\left(1 - \lambda_{\max}\left[\mathbb{E}\left[I - H_B^t\right]\right] - \frac{1}{4}\lambda_{\max}^{H_B}\right)}$$

$$+ \frac{L^2\gamma^2\lambda_{\max}^{H_B}N\sigma^2}{\left(1 - \lambda_{\max}\left[\mathbb{E}\left[I - H_B^t\right]\right] - \frac{1}{4}\lambda_{\max}^{H_B}\right)}.$$

Applying argument from Danilova et al. (2022) we obtain the result for uniformly chosen point. $\quad\square$

**Corollary D.2.1.** *Suppose that Assumption 3.1 and Assumption 5.1 hold. Suppose that a stepsize $\gamma > 0$ is chosen such that $\gamma \leq \frac{1}{2LN}$. Let the updates have a form of several gradient steps (variance $\sigma^2 = 0$) We choose the output of the method $\widetilde{W}^T$ uniformly at random from $W^0, W^1, \ldots, W^{T-1}$ Then, the iterate $\widetilde{W}^T$ of* RAC-LoRA *method (Algorithm 1) with several* GD *updates (Equation 3) satisfy*

$$\mathbb{E}\left[\left\|\nabla f(\widetilde{W}^T)\right\|^2\right] \leq \frac{2}{\gamma NT}\frac{f(W^0) - f^\star}{\left(1 - \lambda_{\max}\left[\mathbb{E}\left[I - H^t\right]\right] - \frac{1}{4}\lambda_{\max}^H\right)}.$$

Given that the step size is divided by the number of gradient steps allocated for each LoRA block, employing multiple gradient steps for a single LoRA block does not provide any significant benefits. This observation suggests that a single gradient step is adequate for each LoRA block. Therefore, in practical applications, it is more advantageous to utilize only one epoch per LoRA block within the training chain. This approach not only streamlines the training process but also optimizes computational efficiency, allowing for more effective resource allocation without compromising the performance of the model.

## D.2 ANALYSIS OF POLYAK-ŁOJASIEWICZ SETTING

Next, we establish the convergence rate for the Polyak-Łojasiewicz setting (Assumption 5.4).

**Theorem D.3.** *Suppose that Assumption 3.1, Assumption 5.4 and Assumption 5.1 hold. Suppose that a stepsize $\gamma \geq 0$ is chosen such that $\gamma \leq \frac{1}{2NL}$. Then, the iterates of* RAC-LoRA *method (Algorithm 1) with* RR *updates (Equation 7) satisfy*

$$\mathbb{E}\left[f(W^T) - f^\star\right] \leq \left(1 - \gamma N \mu \left(1 - \lambda_{\max}\left[\mathbb{E}\left[I - H_B^t\right]\right] - \frac{1}{4}\lambda_{\max}^{H_B}\right)\right)^T \mathbb{E}\left[f(W^0) - f^\star\right]$$
$$+ \frac{L^2\gamma^2\lambda_{\max}^{H_B}N\sigma^2}{2\left(1 - \lambda_{\max}\left[\mathbb{E}\left[I - H_B^t\right]\right] - \frac{1}{4}\lambda_{\max}^{H_B}\right)}.$$

*Proof.* We start from Equation 8:

$$\mathbb{E}\left[f(W^{t+1}) \mid W^t\right] - f^\star \leq f(W^t) - f^\star - \frac{\gamma N}{2}\left\|\nabla f(W^t)\right\|^2 \left(1 - \lambda_{\max}\left[\mathbb{E}\left[I - H_B^t\right]\right] - \frac{1}{4}\lambda_{\max}^{H_B}\right)$$
$$+ \frac{\gamma N}{2}L^2\gamma^2\lambda_{\max}^{H_B}N\sigma^2.$$

Using PL condition $\left\|\nabla f(W^t)\right\|^2 \geq 2\mu\left(f(W^t) - f^\star\right)$ we have

$$\mathbb{E}\left[f(W^{t+1}) \mid W^t\right] - f^\star \leq f(W^t) - f^\star - \gamma N \mu\left(f(W^t) - f^\star\right)\left(1 - \lambda_{\max}\left[\mathbb{E}\left[I - H_B^t\right]\right] - \frac{1}{4}\lambda_{\max}^{H_B}\right)$$
$$+ \frac{\gamma N}{2}L^2\gamma^2\lambda_{\max}^{H_B}N\sigma^2.$$

Taking full expectation we obtain:

$$\mathbb{E}\left[f(W^{t+1}) - f^\star\right] \leq \left(1 - \gamma N \mu\left(1 - \lambda_{\max}\left[\mathbb{E}\left[I - H_B^t\right]\right] - \frac{1}{4}\lambda_{\max}^{H_B}\right)\right)\mathbb{E}\left[f(W^t) - f^\star\right]$$
$$+ \frac{\gamma N}{2}L^2\gamma^2\lambda_{\max}^{H_B}N\sigma^2.$$

After unrolling the recursion we obtain

$$\mathbb{E}\left[f(W^T) - f^\star\right] \leq \left(1 - \gamma N \mu\left(1 - \lambda_{\max}\left[\mathbb{E}\left[I - H_B^t\right]\right] - \frac{1}{4}\lambda_{\max}^{H_B}\right)\right)^T \mathbb{E}\left[f(W^0) - f^\star\right]$$
$$+ \frac{L^2\gamma^2\lambda_{\max}^{H_B}N\sigma^2}{2\left(1 - \lambda_{\max}\left[\mathbb{E}\left[I - H_B^t\right]\right] - \frac{1}{4}\lambda_{\max}^{H_B}\right)}.$$

This finishes the proof. $\qquad\square$

**Corollary D.3.1.** *Suppose that Assumption 3.1, Assumption 5.4 and Assumption 5.1 hold. Let the updates have a form of several gradient steps (variance $\sigma^2 = 0$) Suppose that a stepsize $\gamma \geq 0$ is chosen such that $\gamma \leq \frac{1}{2NL}$. Then, the iterates of* RAC-LoRA *method (Algorithm 1) with several* GD *updates (Equation 3) satisfy*

$$\mathbb{E}\left[f(W^T) - f^\star\right] \leq \left(1 - \gamma N \mu\left(1 - \lambda_{\max}\left[\mathbb{E}\left[I - H_B^t\right]\right] - \frac{1}{4}\lambda_{\max}^{H_B}\right)\right)^T \mathbb{E}\left[f(W^0) - f^\star\right].$$

Since the step size is divided by the number of gradient steps for each LoRA block, using multiple gradient steps does not offer significant advantages. Thus, a single gradient step per LoRA block is sufficient. Practically, it is more efficient to use only one epoch per LoRA block in the training chain.

# E ANALYSIS OF RAC-LoRA WITH SGD UNDER THE ARBITRARY DATA SAMPLING PARADIGM

In the previous section, we introduced the Random Reshuffling (RR) method, where each data point is used exactly once during each epoch, also known as sampling without replacement. This method has demonstrated strong empirical performance across various optimization tasks. However, in this section, we shift our focus to the RAC-LoRA framework, where Stochastic Gradient Descent (SGD) is applied with a more general, arbitrary data sampling procedure, allowing for broader flexibility in how data is selected and used during training.

The analysis of general sampling schemes in SGD has garnered significant attention in the literature, particularly in understanding its impact on convergence rates and optimization performance across different problem classes. For strongly convex functions, general sampling methods have been rigorously studied in works such as Gower et al. (2019), which provide detailed convergence guarantees and bounds. In the case of general convex optimization problems, Khaled et al. (2023) offer a thorough analysis of the performance of SGD under various sampling strategies. Furthermore, for non-convex settings, both Khaled & Richtárik (2020) and Demidovich et al. (2023) have explored how general sampling procedures influence the convergence behavior and optimization efficiency of SGD, shedding light on its applicability to a wide range of machine learning tasks.

In the following sections, we build on these foundational studies to examine how the flexibility of general sampling in the RAC-LoRA framework can lead to improved convergence in certain scenarios, while also maintaining robust performance across different convexity settings.

To conduct this analysis, we introduce a general assumption that extends the standard assumptions presented in Khaled & Richtárik (2020).

The proof is provided for Right Sketch (Definition 4.2). The result for Left Sketch (Definition 4.1) can be derived by following the same steps.

**Assumption E.1** ( Expected smoothness). *The second moment of the stochastic gradient satisfies*

$$\mathbb{E}\left[\|g(W)\|^2\right] \leq 2A_1\left(f(W) - f^{\inf}\right) + B_1 \cdot \|\nabla f(W)\|^2 + C_1$$

*for some $A, B, C \geq 0$ and all $W \in \mathbb{R}^{m \times n}$.*

Now we can also do stochastic analysis. Let us consider the SGD update for LoRA method:

$$\Delta W = \frac{\alpha}{r}\hat{B}A_S,$$

$$W^{t+1} = W^t + \frac{\alpha}{r}\hat{B}^t A_S^t \quad \hat{B}^t = -\gamma g(W^t)(A_S^t)^\top \left(A_S^t(A_S^t)^\top\right)^\dagger$$

Now we have

$$W^{t+1} = W^t - \gamma g(W^t)(A_S^t)^\top \left(A_S^t(A_S^t)^\top\right)^\dagger A_S^t \tag{10}$$
$$= W^t - \gamma g(W^t)H_A^t. \tag{11}$$

## E.1 ANALYSIS OF GENERAL NON-CONVEX SETTING

**Theorem E.2.** *Suppose that Assumption 3.1 and Assumption 5.1 hold. Suppose that a stepsize $\gamma > 0$ is chosen such that $\gamma \leq \min\left[1/\sqrt{LA_1\lambda_{\max}^{H_A}T}, 1/\left(LB_1\frac{\lambda_{\max}^{H_A}}{\lambda_{\min}^{H_A}}\right)\right]$. Then, the iterate $W^T$ of* RAC-LoRA *method (Algorithm 1) with* SGD *updates (Equation 10) satisfy*

$$\min_{0 \leq t \leq T-1} \mathbb{E}\left[\|\nabla f(W^T)\|^2\right] \leq \frac{6}{\lambda_{\min}^{H_A}\gamma T}\left(f(W^0) - f^\star\right) + LC_1\gamma\frac{\lambda_{\max}^{H_A}}{\lambda_{\min}^{H_A}}.$$

*Proof.* We start from $L$-smoothness:

$$f(W^{t+1}) \leq f(W^t) + \left\langle \nabla f(W^t), W^{t+1} - W^t \right\rangle + \frac{L}{2} \left\| W^{t+1} - W^t \right\|^2$$

$$= f(W^t) + \left\langle \nabla f(W^t), -\gamma g(W^t) H_A^t \right\rangle + \frac{L}{2} \left\| -\gamma g(W^t) H_A^t \right\|^2$$

$$= f(W^t) - \gamma \left\langle \nabla f(W^t), g(W^t) H_A^t \right\rangle + \frac{L}{2} \left\| -\gamma g(W^t) H_A^t \right\|^2 .$$

Let us take conditional expectation:

$$\mathbb{E}\left[ f(W^{t+1}) \mid W^t \right] \leq f(W^t) - \gamma \mathbb{E}\left[ \left\langle \nabla f(W^t), g(W^t) H_A^t \right\rangle \mid W^t \right] + \frac{L}{2} \mathbb{E}\left[ \left\| -\gamma g(W^t) H_A^t \right\|^2 \mid W^t \right] .$$

Using that $g(W^t)$ and $H_A^t$ are independent, so we have

$$\mathbb{E}\left[ f(W^{t+1}) \mid W^t \right] \leq f(W^t) - \gamma \left\langle \nabla f(W^t), \mathbb{E}\left[ g(W^t) \right] \mathbb{E}\left[ H_A^t \right] \right\rangle + \frac{L}{2} \mathbb{E}\left[ \left\| -\gamma g(W^t) H_A^t \right\|^2 \mid W^t \right]$$

$$\leq f(W^t) - \gamma \left\langle \nabla f(W^t), \mathbb{E}\left[ g(W^t) \right] \mathbb{E}\left[ H_A^t \right] \right\rangle + \gamma^2 \frac{L}{2} \mathbb{E}\left[ \left\langle g(W^t) H_A^t, g(W^t) H_A^t \right\rangle \mid W^t \right]$$

$$\leq f(W^t) - \gamma \lambda_{\min}\left[ \mathbb{E}\left[ H_A^t \right] \right] \left\| \nabla f(W^t) \right\|^2 + \gamma^2 \frac{L}{2} \mathbb{E}\left[ \left\langle g(W^t) H_A^t, g(W^t) H_A^t \right\rangle \mid W^t \right] .$$

Using the property of projection matrix $H_A^t$, we have

$$\mathbb{E}\left[ f(W^{t+1}) \mid W^t \right] \leq f(W^t) - \gamma \lambda_{\min}\left[ \mathbb{E}\left[ H_A^t \right] \right] \left\| \nabla f(W^t) \right\|^2 + \gamma^2 \frac{L}{2} \lambda_{\max}\left[ \mathbb{E}\left[ H_A^t \right] \right] \mathbb{E}\left[ \left\| g(W^t) \right\|^2 \right] .$$

Now we need to use assumption on stochastic gradients. We will use the most general assumption: ABC – assumption:

$$\mathbb{E}\left[ \| g(W^t) \|^2 \right] \leq 2A_1(f(W^t) - f^\star) + B_1 \left\| \nabla f(W^t) \right\|^2 + C_1 .$$

Now we have

$$\mathbb{E}\left[ f(W^{t+1}) \mid W^t \right] - f^\star \leq f(W^t) - f^\star - \gamma \lambda_{\min}\left[ \mathbb{E}\left[ H_A^t \right] \right] \left\| \nabla f(W^t) \right\|^2$$

$$+ \gamma^2 \frac{L}{2} \lambda_{\max}\left[ \mathbb{E}\left[ H_A^t \right] \right] \left( 2A_1(f(W^t) - f^\star) + B_1 \left\| \nabla f(W^t) \right\|^2 + C_1. \right) .$$

Combining these terms together we get

$$\mathbb{E}\left[ f(W^{t+1}) \mid W^t \right] - f^\star \leq \left( f(W^t) - f^\star \right) \left( 1 + \gamma^2 A_1 L \lambda_{\max}\left[ \mathbb{E}\left[ H_A^t \right] \right] \right) \tag{12}$$

$$- \gamma \lambda_{\min}\left[ \mathbb{E}\left[ H_A^t \right] \right] \left\| \nabla f(W^t) \right\|^2 \left( 1 - \gamma \frac{L}{2} \frac{\lambda_{\max}\left[ \mathbb{E}\left[ H_A^t \right] \right]}{\lambda_{\min}\left[ \mathbb{E}\left[ H_A^t \right] \right]} B_1 \right) \tag{13}$$

$$+ \gamma^2 \frac{L}{2} \lambda_{\max}\left[ \mathbb{E}\left[ H_A^t \right] \right] C_1 . \tag{14}$$

Using condition on stepsize: $1 - \gamma \frac{L B_1}{2} \frac{\lambda_{\max}\left[ \mathbb{E}\left[ H_A^t \right] \right]}{\lambda_{\min}\left[ \mathbb{E}\left[ H_A^t \right] \right]} \geq \frac{1}{2}$ we get

$$\mathbb{E}\left[ f(W^{t+1}) \mid W^t \right] - f^\star \leq \left( f(W^t) - f^\star \right) \left( 1 + \gamma^2 A_1 L \lambda_{\max}\left[ \mathbb{E}\left[ H_A^t \right] \right] \right)$$

$$- \frac{1}{2} \gamma \lambda_{\min}\left[ \mathbb{E}\left[ H_A^t \right] \right] \left\| \nabla f(W^t) \right\|^2$$

$$+ \gamma^2 \frac{L}{2} \lambda_{\max}\left[ \mathbb{E}\left[ H_A^t \right] \right] C_1 .$$

Using tower property of expectation we obtain

$$\mathbb{E}\left[ f(W^{t+1}) - f^\star \right] \leq \mathbb{E}\left[ f(W^t) - f^\star \right] \left( 1 + \gamma^2 A_1 L \lambda_{\max}\left[ \mathbb{E}\left[ H_A^t \right] \right] \right)$$

$$- \frac{1}{2} \gamma \lambda_{\min}\left[ \mathbb{E}\left[ H_A^t \right] \right] \mathbb{E}\left[ \left\| \nabla f(W^t) \right\|^2 \right]$$

$$+ \gamma^2 \frac{L}{2} \lambda_{\max}\left[ \mathbb{E}\left[ H_A^t \right] \right] C_1 .$$

Let us define $\delta^t = \mathbb{E}\left[f(W^t) - f^\star\right]$ and $r^t = \mathbb{E}\left[\left\|\nabla f(W^t)\right\|^2\right]$, after reshuffling of terms we obtain

$$\frac{1}{2}\gamma\lambda_{\min}\left[\mathbb{E}\left[H_A^t\right]\right]\mathbb{E}\left[\left\|\nabla f(W^t)\right\|^2\right] \leq \left(1 + \gamma^2 A_1 L\lambda_{\max}\left[\mathbb{E}\left[H_A^t\right]\right]\right)\delta^t - \delta^{t+1} + \gamma^2\frac{LC_1}{2}\lambda_{\max}\left[\mathbb{E}\left[H_A^t\right]\right].$$

Let use fix $w^{-1} > 0$ and define $w^t = \frac{w^{t-1}}{1 + L\gamma^2 A\lambda_{\max}\left[\mathbb{E}[H_A^t]\right]}$ for all $t \geq 0$. Multiplying by $\frac{w^t}{\gamma}$,

$$\frac{1}{2}w^t r^t \lambda_{\min}\left[\mathbb{E}\left[H_A^t\right]\right] \leq \frac{w^t}{\gamma}\left(1 + \gamma^2 A_1 L\lambda_{\max}\left[\mathbb{E}\left[H_A^t\right]\right]\right)\delta^t - \frac{w^t}{\gamma}\delta^{t+1} + \gamma\frac{LC_1}{2}\lambda_{\max}\left[\mathbb{E}\left[H_A^t\right]\right].$$

Now we obtain

$$\frac{1}{2}w^t r^t \lambda_{\min}\left[\mathbb{E}\left[H_A^t\right]\right] \leq \frac{w^{t-1}}{\gamma}\delta^t - \frac{w^t}{\gamma}\delta^{t+1} + \gamma\frac{LC_1}{2}\lambda_{\max}\left[\mathbb{E}\left[H_A^t\right]\right]w^t.$$

Summing up both sides as $t = 0, 1, \ldots, T-1$ we have,

$$\frac{1}{2}\sum_{t=0}^{T-1}w^t r^t \lambda_{\min}\left[\mathbb{E}\left[H_A^t\right]\right] \leq \frac{w^{-1}}{\gamma}\delta^0 - \frac{w^{T-1}}{\gamma}\delta^T + \gamma\frac{LC_1}{2}\lambda_{\max}\left[\mathbb{E}\left[H_A^t\right]\right]\sum_{t=0}^{T-1}w^t$$

$$\leq \frac{w^{-1}}{\gamma}\delta^0 + \gamma\frac{LC_1}{2}\lambda_{\max}\left[\mathbb{E}\left[H_A^t\right]\right]\sum_{t=0}^{T-1}w^t.$$

Let us define $W^T = \sum_{t=0}^{T-1}w^t$. Dividing both sides by $W^T$ we have,

$$\frac{1}{2}\min_{0\leq t\leq T-1}r^t \leq \frac{1}{W^T}\sum_{t=0}^{T-1}w^t r^t \leq \frac{w^{-1}}{W^T}\frac{\delta^0}{\gamma}\frac{1}{\lambda_{\min}\left[\mathbb{E}\left[H_A^t\right]\right]} + \frac{LC_1\gamma}{2}\frac{\lambda_{\max}\left[\mathbb{E}\left[H_A^t\right]\right]}{\lambda_{\min}\left[\mathbb{E}\left[H_A^t\right]\right]}.$$

Note that,

$$W^T = \sum_{t=0}^{T-1}w^t \geq \sum_{t=0}^{T-1}\min_{0\leq i\leq T-1}w^i = Tw^{T-1} = \frac{Tw^{-1}}{\left(1 + L\gamma^2 A\lambda_{\max}\left[\mathbb{E}\left[H_A^t\right]\right]\right)^T}.$$

Using this we get

$$\frac{1}{2}\min_{0\leq t\leq T-1}r^t \leq \frac{\left(1 + L\gamma^2 A_1\lambda_{\max}\left[\mathbb{E}\left[H_A^t\right]\right]\right)^T}{\lambda_{\min}\left[\mathbb{E}\left[H_A^t\right]\right]\gamma T}\delta^0 + \frac{LC_1\gamma}{2}\frac{\lambda_{\max}\left[\mathbb{E}\left[H_A^t\right]\right]}{\lambda_{\min}\left[\mathbb{E}\left[H_A^t\right]\right]}.$$

Using the fact that $1 + x \leq \exp(x)$, we have that

$$\left(1 + L\gamma^2 A_1\lambda_{\max}\left[\mathbb{E}\left[H_A^t\right]\right]\right)^T \leq \exp\left(L\gamma^2 A_1\lambda_{\max}\left[\mathbb{E}\left[H_A^t\right]\right]T\right) \leq \exp(1) \leq 3$$

where the second inequality holds because $\gamma \leq 1/\sqrt{LA_1\lambda_{\max}\left[\mathbb{E}\left[H_A^t\right]\right]T}$ by assumption. Substituting we get,

$$\min_{0\leq t\leq T-1}r^t \leq \frac{6}{\lambda_{\min}\left[\mathbb{E}\left[H_A^t\right]\right]\gamma T}\left(f(W^0) - f^\star\right) + LC_1\gamma\frac{\lambda_{\max}^{H_A}}{\lambda_{\min}^{H_A}}.$$

$\square$

### E.2 ANALYSIS OF POLYAK-ŁOJASIEWICZ SETTING

In this section we provide analysis of RAC-LoRA method with general SGD update under Polyak-Łojasiewicz condition (Assumption 5.4).

**Theorem E.3.** *Suppose that Assumption 3.1, Assumption 5.4 and Assumption 5.1 hold. Suppose that a stepsize $\gamma \geq 0$ is chosen such that $\gamma \leq \min\left[\frac{\mu}{2A_1 L\frac{\lambda_{\max}^{H_A}}{\lambda_{\min}^{H_A}}}, 1/\left(LB_1\frac{\lambda_{\max}^{H_A}}{\lambda_{\min}^{H_A}}\right)\right]$. Then, the iterates of RAC-LoRA method (Algorithm 1) with SGD updates (Equation 10) satisfy*

$$\mathbb{E}\left[f(W^T)\right] - f^\star \leq \left(1 - \gamma\mu\lambda_{\min}^H\right)^T\left(f(W^0) - f^\star\right).$$

*Proof.* We start from 12:

$$\mathbb{E}\left[f(W^{t+1}) \mid W^t\right] - f^\star \leq \left(f(W^t) - f^\star\right)\left(1 + \gamma^2 A_1 L \lambda_{\max}\left[\mathbb{E}\left[H_A^t\right]\right]\right)$$

$$- \gamma \lambda_{\min}\left[\mathbb{E}\left[H_A^t\right]\right]\left\|\nabla f(W^t)\right\|^2 \left(1 - \gamma \frac{L}{2}\frac{\lambda_{\max}\left[\mathbb{E}\left[H_A^t\right]\right]}{\lambda_{\min}\left[\mathbb{E}\left[H_A^t\right]\right]}B_1\right)$$

$$+ \gamma^2 \frac{L}{2}\lambda_{\max}\left[\mathbb{E}\left[H_A^t\right]\right]C_1.$$

Using $\left(1 - \gamma\frac{L}{2}\frac{\lambda_{\max}\left[\mathbb{E}\left[H_A^t\right]\right]}{\lambda_{\min}\left[\mathbb{E}\left[H_A^t\right]\right]}B_1\right) \geq \frac{3}{4}$ and PL condition we have

$$\mathbb{E}\left[f(W^{t+1}) \mid W^t\right] - f^\star \leq \left(f(W^t) - f^\star\right)\left(1 - \frac{3}{2}\gamma\mu\lambda_{\min}\left[\mathbb{E}\left[H_A^t\right]\right] + \gamma^2 A_1 L\lambda_{\max}\left[\mathbb{E}\left[H_A^t\right]\right]\right)$$

$$+ \gamma^2 \frac{L}{2}\lambda_{\max}\left[\mathbb{E}\left[H_A^t\right]\right]C_1.$$

Using that $LA_1\gamma\lambda_{\max}\left[\mathbb{E}\left[H_A^t\right]\right] \leq \frac{\mu}{2}\lambda_{\min}\left[\mathbb{E}\left[H_A^t\right]\right]$ we obtain

$$\mathbb{E}\left[f(W^{t+1}) \mid W^t\right] - f^\star \leq \left(f(W^t) - f^\star\right)\left(1 - \gamma\mu\lambda_{\min}\left[\mathbb{E}\left[H_A^t\right]\right]\right) + \gamma^2 \frac{L}{2}\lambda_{\max}\left[\mathbb{E}\left[H_A^t\right]\right]C_1.$$

Taking full expectation and using tower property we obtain:

$$\mathbb{E}\left[f(W^{t+1}) - f^\star\right] \leq \mathbb{E}\left[f(W^t) - f^\star\right]\left(1 - \gamma\mu\lambda_{\min}\left[\mathbb{E}\left[H_A^t\right]\right]\right) + \gamma^2 \frac{L}{2}\lambda_{\max}\left[\mathbb{E}\left[H_A^t\right]\right]C_1.$$

Once we unroll the recursion we obtain

$$\mathbb{E}\left[f(W^T) - f^\star\right] \leq \mathbb{E}\left[f(W^0) - f^\star\right]\left(1 - \gamma\mu\lambda_{\min}^{H_A}\right)^T + \gamma\frac{L}{2\mu\lambda_{\min}^{H_A}}\lambda_{\max}^{H_A}C_1.$$

$\square$

---

**Algorithm 2** Federated Randomized Asymmetric Chain of LoRA (Fed-RAC-LoRA)

---

1: **Parameters:** initial pre-trained model $W^0$, rank $r$, learning rate $\gamma > 0$, scaling factor $\alpha$, server stepsize $\beta > 0$ number of modules in chain $T$, sample distribution $\mathcal{D}_S^B$ or $\mathcal{D}_S^A$.

2: **for** $t = 0, 1, \ldots, T - 1$ **do**

3:     Sample a subset (cohort) of clients $S^t$

4:         (Option 1)    Sample a matrix $B_S^t$          (Option 2)    Sample a matrix $A_S^t$

5:     Send the model $W^t$ and fixed matrix (Option 1) $B_S^t$ or (Option 2) $A_S^t$ to clients

6:     **for** $m \in S^t$ **do**

7:         Solve subproblem

$$\text{(Option 1) } \hat{A}_m^t \approx \min_A f_m(W^t + \frac{\alpha}{r} B_S^t A) \qquad \text{(Option 2) } \hat{B}_m^t \approx \min_B f_m(W^t + \frac{\alpha}{r} B A_S^t)$$

8:         Send the updates to server (Option 1) $\hat{A}_m^t$ or (Option 2) $\hat{B}_m^t$

9:     **end for**

10:    Merge the updates

11:

$$\text{(Option 1)} \quad W^{t+1} = W^t + \beta \frac{\alpha}{r} B_S^t \frac{1}{C} \sum_{m \in S^t} \hat{A}_m^t$$

12:

$$\text{(Option 2)} \quad W^{t+1} = W^t + \beta \frac{\alpha}{r} \frac{1}{C} \sum_{m \in S^t} \hat{B}_m^t A_S^t$$

13: **end for**

---

## F    FEDERATED LEARNING SETTING

We consider the following optimization problem with a double finite-sum structure:

$$\min_{\Delta W \in \mathbb{R}^{m \times n}} f(W^0 + \Delta W) = \frac{1}{M} \sum_{m=1}^{M} \frac{1}{N} \sum_{i=1}^{N} f_{m,i}(W^0 + \Delta W), \tag{15}$$

where $M$ is the total number of clients and $N$ is the number of data points on each client. In the context of Federated Learning, each client maintains its own local loss function $f_m$, which also follows a finite-sum structure, reflecting the client's local data. This formulation captures the decentralized nature of the learning process, where each client performs computations based on their local dataset.

Federated Learning (FL) (Konečný et al., 2016; Kairouz et al., 2021) is a distributed machine learning framework that enables multiple devices or clients to collaboratively train a shared model without sending their raw data to a central server. In contrast to traditional machine learning, where data is centralized for model training, Federated Learning allows each client to train a local model using its own data. The clients then share only the updated model parameters with a central server or aggregator. The server aggregates these updates to form a new global model, which is then redistributed to the clients for further iterations of the process (Konečný et al., 2016). Local Training (LT) is a key component of Federated Learning (FL), in which each participating client conducts several local optimization steps before synchronizing their model parameters with the central server.

The analysis of LT marked a significant advancement by eliminating the need for data homogeneity assumptions, as demonstrated by Khaled et al. (2019; 2020). However, later studies by Woodworth et al. (2020b) and Glasgow et al. (2022) revealed that LocalSGD (also known as FedAvg) has no communication complexity advantage over minibatch SGD in heterogeneous data settings. Additionally, Malinovskiy et al. (2020) analyzed LT methods for general fixed-point problems, while Koloskova et al. (2020) explored decentralized aspects of LT.

Although removing the data homogeneity requirement was a major breakthrough, the results were somewhat discouraging, as they indicated that LT-enhanced GD, or LocalGD, exhibits a sublinear convergence rate, which is worse than the linear convergence rate of vanilla GD (Woodworth et al.,

2020a). The impact of server-side step sizes was further explored by Malinovsky et al. (2023b) and Charles & Konečný (2020).

Subsequent LT methods aimed to achieve linear convergence by addressing client drift, which had hindered earlier approaches. Scaffold, introduced by Karimireddy et al. (2020), was the first to successfully mitigate client drift and achieve a linear convergence rate. Similar methods were later proposed by Gorbunov et al. (2021). Although this was a significant breakthrough, these methods still have slightly higher or equal communication complexity compared to vanilla GD.

Mishchenko et al. (2022b) recently introduced the ProxSkip method, a simple yet effective approach to Local Training that achieves provable communication acceleration in the smooth strongly convex regime, even with heterogeneous data. In a follow-up article, Malinovsky et al. (2022) expanded on ProxSkip, presenting a broad variance reduction framework. Condat & Richtárik (2022) further applied ProxSkip to complex splitting schemes involving the sum of three operators in a forward-backward setting. Additionally, Sadiev et al. (2022a) and Maranjyan et al. (2022) improved the computational complexity of ProxSkip while preserving its communication efficiency. Condat et al. (2023) introduced accelerated Local Training methods allowing client sampling based on ProxSkip, while Grudzień et al. (2023a;b) proposed an accelerated method using the RandProx approach with primal and dual updates.

In practice, Federated Learning faces a fundamental challenge: it is often infeasible for all clients to communicate and aggregate updates with the central server simultaneously due to limitations such as network bandwidth, client availability, or resource constraints. Therefore, rather than requiring all clients to participate in every round of communication, we adopt a strategy in which only a randomly selected subset of clients is involved in each aggregation step. This approach relies on uniform sampling of the clients, ensuring that the selection process is unbiased over time.

The method operates as follows: in each communication round, the central server sends the current global model, denoted by $W^t$, along with the sampled matrix, to the clients chosen to participate in the current cohort. Each client in this cohort trains a local learnable matrix using an optimization algorithm (e.g., stochastic gradient descent) based on their local data. After completing the local updates, the clients send their computed updates (i.e., changes in model parameters) back to the central server.

Once the server receives these updates, it aggregates them (e.g., by averaging the updates) to produce an updated global model. In addition to the aggregation, the server may perform an additional server-side update step to further refine the model before broadcasting it in the next round. This iterative process of local training, communication, and aggregation continues until convergence is achieved or a predefined stopping criterion is met.

The proof is provided for Left Sketch (Definition 4.1). The result for Right Sketch (Definition 4.2) can be derived by following the same steps.

For local optimzier we use Random Reshuffling, where the effective step has a form:

$$W_{m,i+1}^t = W_{m,i}^t - \gamma H_B^t \nabla f_{m,i}(W_{m,i}^t) \tag{16}$$

The server-side step looks like $W^{t+1} = W^t - \tilde{\eta} H_B^t \frac{1}{C} \sum_{m \in S^t} \hat{A}_m^t$.

Let us formulate nesesary assumptions

**Assumption F.1** (Functional dissimilarity). *The variance at the optimum in the non-convex regime is defined as*

$$\Delta^\star \overset{def}{=} f^\star - \frac{1}{M} \sum_{m=1}^M f_m^\star$$

*where $f_m^\star = \inf_W f_m(W)$ and $f^\star = \inf_W f(W)$. For each device $m$, the variance at the optimum is defined as*

$$\Delta_m^\star \overset{def}{=} f^\star - \frac{1}{n} \sum_{i=1}^n f_{m,i}^\star$$

*where $f_{m,i}^\star = \inf_W f_{m,i}(W)$*

## F.1 ANALYSIS OF GENERAL NON-CONVEX SETTING

**Theorem F.2.** *Suppose that Assumption 3.1 and Assumption 5.1 hold. Suppose that stepsizes $\gamma, \tilde{\eta} > 0$ is chosen such that $\gamma n \leq \tilde{\eta} \leq \frac{1 - \lambda_{\min}^{H_B}}{4L}$. Then, the iterate $W^T$ of* Fed-RAC-LoRA *method (Algorithm 2) with* RR *updates (Equation 16) satisfy*

$$
\min_{t=0,\ldots,T-1} \mathbb{E}\left[\left\|\nabla f\left(W^t\right)\right\|^2\right] \leq \frac{4\left(1 + 4\tilde{\eta}L^3\gamma^2N^2 + 2L^2\tilde{\eta}^2\frac{M-C}{C\max\{M-1,1\}}\right)^T}{\lambda_{\max}\left[\mathbb{E}\left[I - H_B^t\right]\right]\tilde{\eta}T}\left(f(W^0) - f^\star\right)
$$

$$
+ \frac{8\gamma^2NL^3}{\lambda_{\max}\left[\mathbb{E}\left[I - H_B^t\right]\right]}\left(\frac{1}{M}\sum_{m=1}^M\Delta_m^* + N\Delta^*\right)
$$

$$
+ \frac{8L^2\tilde{\eta}}{\lambda_{\max}\left[\mathbb{E}\left[I - H_B^t\right]\right]}\frac{M-C}{C\max\{M-1,1\}}\Delta^*.
$$

*Proof.* We start from $L$-smoothness:

$$
f(W^{t+1}) \leq f(W^t) + \left\langle \nabla f(W^t), W^{t+1} - W^t \right\rangle + \frac{L}{2}\left\|W^{t+1} - W^t\right\|^2
$$

$$
\leq f(W^t) - \left\langle \nabla f(W^t), \tilde{\eta}\frac{1}{CN}\sum_{m\in S^t}\sum_{i=0}^{N-1}H^t\nabla f_m^{\pi_{m,i}^t}\left(W_{m,i}^t\right) \right\rangle
$$

$$
+ \frac{L}{2}\left\|\tilde{\eta}\frac{1}{CN}\sum_{m\in S^t}\sum_{i=0}^{N-1}H^t\nabla f_m^{\pi_{m,i}^t}(W_{m,i}^t)\right\|^2.
$$

Now we take expectation with respect to sampling:

$$
\mathbb{E}_{S^t}\left[f(W^{t+1})\right] \leq f(W^t) - \tilde{\eta}\mathbb{E}_{S^t}\left[\left\langle \nabla f(W^t), \frac{1}{CN}\sum_{m\in S^t}\sum_{i=0}^{N-1}H_B^t\nabla f_m^{\pi_{m,i}^t}\left(W_{m,i}^t\right) \right\rangle\right]
$$

$$
+ \frac{L}{2}\tilde{\eta}^2\mathbb{E}_{S^t}\left[\left\|\frac{1}{CN}\sum_{m\in S^t}\sum_{i=0}^{N-1}H_B^t\nabla f_m^{\pi_{m,i}^t}(W_{m,i}^t)\right\|^2\right]
$$

$$
\leq f(W^t) - \tilde{\eta}\left\langle \nabla f(W^t), \frac{1}{MN}\sum_{m=1}^M\sum_{i=0}^{N-1}H_B^t\nabla f_m^{\pi_{m,i}^t}\left(W_{m,i}^t\right) \right\rangle
$$

$$
+ \frac{L}{2}\tilde{\eta}^2\mathbb{E}_{S^t}\left[\left\|\frac{1}{CN}\sum_{m\in S^t}\sum_{i=0}^{N-1}H_B^t\nabla f_m^{\pi_{m,i}^t}(W_{m,i}^t)\right\|^2\right].
$$

Using $2\langle a, b\rangle = \|a+b\|^2 - \|a\|^2 - \|b\|^2$, we have

$$
\mathbb{E}_{S^t}\left[f(W^{t+1})\right] \leq f(W^t) - \frac{\tilde{\eta}}{2}\|\nabla f(W^t)\|^2 - \frac{\tilde{\eta}}{2}\left\|\frac{1}{MN}\sum_{m=1}^{M}\sum_{i=0}^{N-1} H_B^t \nabla f_m^{\pi_{m,i}^t}(W_{m,i}^t)\right\|^2
$$

$$
+ \frac{\tilde{\eta}}{2}\left\|\nabla f(W^t) - \frac{1}{MN}\sum_{m=1}^{M}\sum_{i=0}^{N-1} H_B^t \nabla f_m^{\pi_{m,i}^t}(W_{m,i}^t)\right\|^2
$$

$$
+ \frac{L}{2}\tilde{\eta}^2 \mathbb{E}_{S^t}\left[\left\|\frac{1}{CN}\sum_{m\in S^t}\sum_{i=0}^{N-1} H_B^t \nabla f_m^{\pi_{m,i}^t}(W_{m,i}^t)\right\|^2\right]
$$

$$
\leq f(W^t) - \frac{\tilde{\eta}}{2}\|\nabla f(W^t)\|^2 + \frac{\tilde{\eta}}{2}\left\|\nabla f(W^t) - \frac{1}{MN}\sum_{m=1}^{M}\sum_{i=0}^{N-1} H_B^t \nabla f_m^{\pi_{m,i}^t}(W_{m,i}^t)\right\|^2
$$

$$
+ \frac{L}{2}\tilde{\eta}^2 \mathbb{E}_{S^t}\left[\left\|\frac{1}{CN}\sum_{m\in S^t}\sum_{i=0}^{N-1} H_B^t \nabla f_m^{\pi_{m,i}^t}(W_{m,i}^t)\right\|^2\right].
$$

Now we need to add and subtract $H_B^t \nabla f(W^t)$:

$$
\mathbb{E}_{S^t}\left[f(W^{t+1})\right] \leq f(W^t) - \frac{\tilde{\eta}}{2}\|\nabla f(W^t)\|^2
$$

$$
+ \frac{\tilde{\eta}}{2}\left\|\nabla f(W^t) - H_B^t \nabla f(W^t) + H_B^t \nabla f(W^t) - \frac{1}{MN}\sum_{m=1}^{M}\sum_{i=0}^{N-1} H_B^t \nabla f_m^{\pi_{m,i}^t}(W_{m,i}^t)\right\|^2
$$

$$
+ \frac{L}{2}\tilde{\eta}^2 \mathbb{E}_{S^t}\left[\left\|\frac{1}{CN}\sum_{m\in S^t}\sum_{i=0}^{N-1} H_B^t \nabla f_m^{\pi_{m,i}^t}(W_{m,i}^t)\right\|^2\right]
$$

$$
\leq f(W^t) - \frac{\tilde{\eta}}{2}\|\nabla f(W^t)\|^2
$$

$$
+ \frac{\tilde{\eta}}{2}\left\|\nabla f(W^t)\left(I - H_B^t\right) + \frac{1}{MN}\sum_{m=1}^{M}\sum_{i=0}^{N-1} H_B^t \nabla f_m^{\pi_{m,i}^t}(W^t) - \frac{1}{MN}\sum_{m=1}^{M}\sum_{i=0}^{N-1} H_B^t \nabla f_m^{\pi_{m,i}^t}(W_{m,i}^t)\right\|^2
$$

$$
+ \frac{L}{2}\tilde{\eta}^2 \mathbb{E}_{S^t}\left[\left\|\frac{1}{CN}\sum_{m\in S^t}\sum_{i=0}^{N-1} H_B^t \nabla f_m^{\pi_{m,i}^t}(W_{m,i}^t)\right\|^2\right]
$$

$$
\leq f(W^t) - \frac{\tilde{\eta}}{2}\|\nabla f(W^t)\|^2
$$

$$
+ \frac{\tilde{\eta}}{2}\left\|\nabla f(W^t)\left(I - H_B^t\right) + \frac{1}{MN}\sum_{m=1}^{M}\sum_{i=0}^{N-1} H_B^t \left(\nabla f_m^{\pi_{m,i}^t}(W^t) - \nabla f_m^{\pi_{m,i}^t}(W_{m,i}^t)\right)\right\|^2
$$

$$
+ \frac{L}{2}\tilde{\eta}^2 \mathbb{E}_{S^t}\left[\left\|\frac{1}{CN}\sum_{m\in S^t}\sum_{i=0}^{N-1} H_B^t \nabla f_m^{\pi_{m,i}^t}(W_{m,i}^t)\right\|^2\right].
$$

Since $H_B^t(I - H_B^t) = 0$ we obtain

$$
\mathbb{E}_{S^t}\left[f(W^{t+1})\right] \leq f(W^t) - \frac{\tilde{\eta}}{2}\|\nabla f(W^t)\|^2
$$

$$
+ \frac{\tilde{\eta}}{2}\left\|\nabla f(W^t)\left(I - H_B^t\right)\right\|^2 + \frac{\tilde{\eta}}{2}\left\|\frac{1}{MN}\sum_{m=1}^{M}\sum_{i=0}^{N-1} H_B^t \left(\nabla f_m^{\pi_{m,i}^t}(W^t) - \nabla f_m^{\pi_{m,i}^t}(W_{m,i}^t)\right)\right\|^2
$$

$$
+ \frac{L}{2}\tilde{\eta}^2 \mathbb{E}_{S^t}\left[\left\|\frac{1}{CN}\sum_{m\in S^t}\sum_{i=0}^{N-1} H_B^t \nabla f_m^{\pi_{m,i}^t}(W_{m,i}^t)\right\|^2\right].
$$

Now we take conditional expectation and use tower property:

$$
\begin{aligned}
\mathbb{E}\left[f(W^{t+1}) \mid W^t\right] \leq & f(W^t) - \frac{\tilde{\eta}}{2}\left\|\nabla f(W^t)\right\|^2 + \frac{\tilde{\eta}}{2}\mathbb{E}\left[\left\|\nabla f(W^t)\left(I - H_B^t\right)\right\|^2 \mid W^t\right] \\
& + \frac{\tilde{\eta}}{2}\mathbb{E}\left[\left\|\frac{1}{MN}\sum_{m=1}^{M}\sum_{i=0}^{N-1}H_B^t\left(\nabla f_m^{\pi_{m,i}^t}(W^t) - \nabla f_m^{\pi_{m,i}^t}(W_{m,i}^t)\right)\right\|^2 \mid W^t\right] \\
& + \frac{L}{2}\tilde{\eta}^2\mathbb{E}\left[\mathbb{E}_{S^t}\left[\left\|\frac{1}{CN}\sum_{m\in S^t}\sum_{i=0}^{N-1}H_B^t\nabla f_m^{\pi_{m,i}^t}(W_{m,i}^t)\right\|^2\right] \mid W^t\right].
\end{aligned}
$$

Next, we use eigenvalues to obtain bounds:

$$
\begin{aligned}
\mathbb{E}\left[f(W^{t+1}) \mid W^t\right] \leq & f(W^t) - \frac{\tilde{\eta}}{2}\left\|\nabla f(W^t)\right\|^2 + \frac{\tilde{\eta}}{2}\lambda_{\max}\left[\mathbb{E}\left[I - H_B^t\right]\right]\|\nabla f(W^t)\|^2 \\
& + \frac{\tilde{\eta}}{2}\mathbb{E}\left[\lambda_{\max}\left[H_B^t\right]\left\|\frac{1}{MN}\sum_{m=1}^{M}\sum_{i=0}^{N-1}\left(\nabla f_m^{\pi_{m,i}^t}(W^t) - \nabla f_m^{\pi_{m,i}^t}(W_{m,i}^t)\right)\right\|^2 \mid W^t\right] \\
& + \frac{L}{2}\tilde{\eta}^2\mathbb{E}\left[\mathbb{E}_{S^t}\left[\lambda_{\max}\left[H_B^t\right]\left\|\frac{1}{Cn}\sum_{m\in S^t}\sum_{i=0}^{N-1}\nabla f_m^{\pi_{m,i}^t}(W_{m,i}^t)\right\|^2\right] \mid W^t\right].
\end{aligned}
$$

Since $\lambda_{\max}\left[H_B^t\right] = 1$ we have

$$
\begin{aligned}
\mathbb{E}\left[f(W^{t+1}) \mid W^t\right] \leq & f(W^t) - \frac{\tilde{\eta}}{2}\left\|\nabla f(W^t)\right\|^2 + \frac{\tilde{\eta}}{2}\lambda_{\max}\left[\mathbb{E}\left[I - H_B^t\right]\right]\|\nabla f(W^t)\|^2 \\
& + \frac{\tilde{\eta}}{2}\mathbb{E}\left[\left\|\frac{1}{MN}\sum_{m=1}^{M}\sum_{i=0}^{N-1}\left(\nabla f_m^{\pi_{m,i}^t}(W^t) - \nabla f_m^{\pi_{m,i}^t}(W_{m,i}^t)\right)\right\|^2 \mid W^t\right] \\
& + \frac{L}{2}\tilde{\eta}^2\mathbb{E}\left[\mathbb{E}_{S^t}\left[\left\|\frac{1}{CN}\sum_{m\in S^t}\sum_{i=0}^{N-1}\nabla f_m^{\pi_{m,i}^t}(W_{m,i}^t)\right\|^2\right] \mid W^t\right].
\end{aligned}
$$

Using Lemma 5 from Malinovsky et al. (2023b) we have

$$
\begin{aligned}
& \frac{L}{2}\tilde{\eta}^2\mathbb{E}\left[\mathbb{E}_{S^t}\left[\left\|\frac{1}{CN}\sum_{m\in S^t}\sum_{i=0}^{N-1}\nabla f_m^{\pi_{m,i}^t}(W_{m,i}^t)\right\|^2\right] \mid W^t\right] \\
& \leq L^3\tilde{\eta}^2\mathbb{E}\left[\frac{1}{Mn}\sum_{m=1}^{M}\sum_{i=0}^{N-1}\left\|W_{m,i}^t - W^t\right\|^2 \mid W^t\right] \\
& \quad + L\tilde{\eta}^2\left\|\nabla f\left(W^t\right)\right\|^2 \\
& \quad + L\tilde{\eta}^2\frac{M - C}{C\max\{M-1,1\}}\left(2L\left(f\left(W^t\right) - f^\star\right) + 2L\Delta^\star\right)
\end{aligned}
$$

Using this bound and $L$-smoothness for the term in second line we obtain:

$$
\begin{aligned}
\mathbb{E}\left[f(W^{t+1}) \mid W^t\right] \leq & f(W^t) - \frac{\tilde{\eta}}{2}\left\|\nabla f(W^t)\right\|^2 + \frac{\tilde{\eta}}{2}\lambda_{\max}\left[\mathbb{E}\left[I - H_B^t\right]\right]\|\nabla f(W^t)\|^2 \\
& + \frac{\tilde{\eta}}{2}L^2\mathbb{E}\left[\frac{1}{Mn}\sum_{m=1}^{M}\sum_{i=0}^{n-1}\left\|W^t - W_{m,i}^t\right\|^2 \mid W^t\right] \\
& + L^3\tilde{\eta}^2\mathbb{E}\left[\frac{1}{Mn}\sum_{m=1}^{M}\sum_{i=0}^{n-1}\left\|W^t - W_{m,i}^t\right\|^2 \mid W^t\right] + L\tilde{\eta}^2\left\|\nabla f\left(W^t\right)\right\|^2 \\
& + L\tilde{\eta}^2\frac{M - C}{C\max\{M-1,1\}}\left(2L\left(f\left(W^t\right) - f^\star\right) + 2L\Delta^\star\right)
\end{aligned}
$$

Since $\tilde{\eta} \leq \frac{1}{2L}$ we get

$$\mathbb{E}\left[f(W^{t+1}) \mid W^t\right] \leq f(W^t) - \frac{\tilde{\eta}}{2}\left\|\nabla f(W^t)\right\|^2 + \frac{\tilde{\eta}}{2}\lambda_{\max}\left[\mathbb{E}\left[I - H_B^t\right]\right]\left\|\nabla f(W^t)\right\|^2$$

$$+ \tilde{\eta}L^2\mathbb{E}\left[\frac{1}{MN}\sum_{m=1}^{M}\sum_{i=0}^{N-1}\left\|W^t - W_{m,i}^t\right\|^2 \mid W^t\right] + L\tilde{\eta}^2\left\|\nabla f\left(W^t\right)\right\|^2$$

$$+ L\tilde{\eta}^2\frac{M-C}{C\max\{M-1,1\}}\left(2L\left(f\left(W^t\right) - f^\star\right) + 2L\Delta^\star\right).$$

Using lemma 6 from (cite) we obtain

$$\frac{1}{MN}\sum_{m=1}^{M}\sum_{i=0}^{N-1}\mathbb{E}\left[\left\|W^t - W_{m,i}^t\right\|^2 \mid W^t\right] \leq 4\gamma^2N^2L\left(f\left(W^t\right) - f^*\right)$$

$$+ 2\gamma^2N^2L\Delta^* + 2\gamma^2NL\frac{1}{M}\sum_{m=1}^{M}\Delta_m^*.$$

Plugging this bound we obtain

$$\mathbb{E}\left[f(W^{t+1}) \mid W^t\right] \leq f(W^t) - \frac{\tilde{\eta}}{2}\left\|\nabla f(W^t)\right\|^2 + \frac{\tilde{\eta}}{2}\lambda_{\max}\left[\mathbb{E}\left[I - H_B^t\right]\right]\left\|\nabla f(W^t)\right\|^2$$

$$+ \tilde{\eta}L^2\left(4\gamma^2N^2L\left(f\left(W^t\right) - f^*\right) + 2\gamma^2N^2L\Delta^* + 2\gamma^2NL\frac{1}{M}\sum_{m=1}^{M}\Delta_m^*\right)$$

$$+ L\tilde{\eta}^2\left\|\nabla f\left(W^t\right)\right\|^2$$

$$+ L\tilde{\eta}^2\frac{M-C}{C\max\{M-1,1\}}\left(2L\left(f\left(W^t\right) - f^\star\right) + 2L\Delta^\star\right).$$

Next, we have

$$\mathbb{E}\left[f(W^{t+1}) \mid W^t\right] \leq f(W^t) - \frac{\tilde{\eta}}{2}\left\|\nabla f(W^t)\right\|^2\left(1 - \lambda_{\max}\left[\mathbb{E}\left[I - H_B^t\right]\right] - 2L\tilde{\eta}\right)$$

$$+ \tilde{\eta}L^2\left(4\gamma^2N^2L\left(f\left(W^t\right) - f^*\right) + 2\gamma^2N^2L\Delta^* + 2\gamma^2NL\frac{1}{M}\sum_{m=1}^{M}\Delta_m^*\right)$$

$$+ L\tilde{\eta}^2\frac{M-C}{C\max\{M-1,1\}}\left(2L\left(f\left(W^t\right) - f^\star\right) + 2L\Delta^\star\right).$$

Using $\tilde{\eta} \leq \frac{1-\lambda_{\max}\left[\mathbb{E}\left[H_B^t\right]\right]}{4L}$ we get

$$\mathbb{E}\left[f(W^{t+1}) \mid W^t\right] \leq f(W^t) - \frac{\tilde{\eta}}{4}\left\|\nabla f(W^t)\right\|^2\left(1 - \lambda_{\max}\left[\mathbb{E}\left[I - H_B^t\right]\right]\right)$$

$$+ \tilde{\eta}L^2\left(4\gamma^2N^2L\left(f\left(W^t\right) - f^*\right) + 2\gamma^2N^2L\Delta^* + 2\gamma^2NL\frac{1}{M}\sum_{m=1}^{M}\Delta_m^*\right)$$

$$+ L\tilde{\eta}^2\frac{M-C}{C\max\{M-1,1\}}\left(2L\left(f\left(W^t\right) - f^\star\right) + 2L\Delta^\star\right).$$

Next, we subtract $f^\star$ from both sides:

$$\mathbb{E}\left[f(W^{t+1}) \mid W^t\right] - f^\star \leq f(W^t) - f^\star - \frac{\tilde{\eta}}{4}\left\|\nabla f(W^t)\right\|^2\left(1 - \lambda_{\max}\left[\mathbb{E}\left[I - H_B^t\right]\right]\right)$$

$$+ \tilde{\eta}L^2\left(4\gamma^2N^2L\left(f\left(W^t\right) - f^*\right) + 2\gamma^2N^2L\Delta^* + 2\gamma^2NL\frac{1}{M}\sum_{m=1}^{M}\Delta_m^*\right)$$

$$+ L\tilde{\eta}^2\frac{M-C}{C\max\{M-1,1\}}\left(2L\left(f\left(W^t\right) - f^\star\right) + 2L\Delta^\star\right).$$

Taking full expectation we obtain

$$\mathbb{E}\left[f(W^{t+1}) - f^\star\right] \leq \mathbb{E}\left[f(W^t) - f^\star\right]\left(1 + 4\tilde{\eta}L^3\gamma^2 N^2 + 2L^2\tilde{\eta}^2\frac{M - C}{C\max\{M-1,1\}}\right)$$

$$- \frac{\tilde{\eta}}{4}\left\|\nabla f(W^t)\right\|^2\left(1 - \lambda_{\max}\left[\mathbb{E}\left[I - H_B^t\right]\right]\right)$$

$$+ \tilde{\eta}L^2\left(2\gamma^2 N^2 L\Delta^* + 2\gamma^2 NL\frac{1}{M}\sum_{m=1}^M \Delta_m^*\right) + 2L^2\tilde{\eta}^2\frac{M - C}{C\max\{M-1,1\}}\Delta^*.$$

Next, we apply lemma from Khaled & Richtárik (2020) and obtain

$$\min_{t=0,\dots,T-1}\mathbb{E}\left[\left\|\nabla f\left(W^t\right)\right\|^2\right] \leq \frac{4\left(1 + 4\tilde{\eta}L^3\gamma^2 N^2 + 2L^2\tilde{\eta}^2\frac{M-C}{C\max\{M-1,1\}}\right)^T}{\lambda_{\max}\left[\mathbb{E}\left[I - H_B^t\right]\right]\tilde{\eta}T}\left(f(W^0) - f^\star\right)$$

$$+ \frac{8\gamma^2 NL^3}{\lambda_{\max}\left[\mathbb{E}\left[I - H_B^t\right]\right]}\left(\frac{1}{M}\sum_{m=1}^M \Delta_m^* + N\Delta^*\right)$$

$$+ \frac{8L^2\tilde{\eta}}{\lambda_{\max}\left[\mathbb{E}\left[I - H_B^t\right]\right]}\frac{M - C}{C\max\{M-1,1\}}\Delta^*.$$

$$\square$$

## F.2 ANALYSIS OF POLYAK-ŁOJASIEWICZ SETTING

**Theorem F.3.** *Suppose that Assumption 3.1, Assumption 5.4 and Assumption 5.1 hold. Suppose that stepsizes $\gamma, \tilde{\eta} > 0$ is chosen such that $\gamma n \leq \tilde{\eta} \leq \frac{1-\lambda_{\min}^{H_B}}{4L}$. Then, the iterate $W^T$ of* Fed-RAC-LoRA *method (Algorithm 2) with* RR *updates (Equation 16) satisfy*

$$\mathbb{E}\left[f(W^{t+1}) - f^\star\right] \leq \left(f(W^0) - f^\star\right)\left(1 - \tilde{\eta}\mu\left(1 - \lambda_{\max}\left[\mathbb{E}\left[I - H^t\right]\right] - 3L\tilde{\eta}\right)\right)^T$$

$$+ \frac{\tilde{\eta}L^2}{\tilde{\eta}\mu\left(1 - \lambda_{\max}\left[\mathbb{E}\left[I - H^t\right]\right] - 3L\tilde{\eta}\right)}\left(2\gamma^2 N^2 L\Delta^* + 2\gamma^2 NL\frac{1}{M}\sum_{m=1}^M \Delta_m^*\right)$$

$$+ \frac{L\tilde{\eta}^2}{\tilde{\eta}\mu\left(1 - \lambda_{\max}\left[\mathbb{E}\left[I - H^t\right]\right] - 3L\tilde{\eta}\right)}\frac{M - C}{C\max\{M-1,1\}}\left(2L\left(f\left(W^t\right) - f^\star\right) + 2L\Delta^*\right).$$

*Proof.* We start from

$$\mathbb{E}\left[f(W^{t+1}) \mid W^t\right] \leq f(W^t) - \frac{\tilde{\eta}}{2}\left\|\nabla f(W^t)\right\|^2\left(1 - \lambda_{\max}\left[\mathbb{E}\left[I - H^t\right]\right] - 2L\tilde{\eta}\right)$$

$$+ \tilde{\eta}L^2\left(4\gamma^2 N^2 L\left(f\left(W^t\right) - f^*\right) + 2\gamma^2 N^2 L\Delta^* + 2\gamma^2 NL\frac{1}{M}\sum_{m=1}^M \Delta_m^*\right)$$

$$+ L\tilde{\eta}^2\frac{M - C}{C\max\{M-1,1\}}\left(2L\left(f\left(W^t\right) - f^\star\right) + 2L\Delta^*\right).$$

Using Assumption5.4 we have

$$\mathbb{E}\left[f(W^{t+1}) \mid W^t\right] \leq f(W^t) - \tilde{\eta}\mu\left\|\nabla f(W^t)\right\|^2\left(1 - \lambda_{\max}\left[\mathbb{E}\left[I - H^t\right]\right] - 2L\tilde{\eta}\right)\left(f(W^t) - f^\star\right)$$

$$+ \tilde{\eta}L^2\left(4\gamma^2 n^2 L\left(f\left(W^t\right) - f^*\right) + 2\gamma^2 N^2 L\Delta^* + 2\gamma^2 NL\frac{1}{M}\sum_{m=1}^M \Delta_m^*\right)$$

$$+ L\tilde{\eta}^2\frac{M - C}{C\max\{M-1,1\}}\left(2L\left(f\left(W^t\right) - f^\star\right) + 2L\Delta^*\right).$$

Using the stepsize $\gamma \leq \frac{1}{4nL}$ we have

$$\mathbb{E}\left[f(W^{t+1}) \mid W^t\right] \leq f(W^t) - \tilde{\eta}\mu\left(1 - \lambda_{\max}\left[\mathbb{E}\left[I - H^t\right]\right] - 3L\tilde{\eta}\right)\left(f(W^t) - f^\star\right)$$

$$+ \tilde{\eta}L^2\left(2\gamma^2 N^2 L\Delta^* + 2\gamma^2 NL\frac{1}{M}\sum_{m=1}^{M}\Delta_m^*\right)$$

$$+ L\tilde{\eta}^2\frac{M-C}{C\max\{M-1,1\}}\left(2L\left(f\left(W^t\right) - f^\star\right) + 2L\Delta^*\right).$$

After unrolling the recursion we obtain

$$\mathbb{E}\left[f(W^{t+1}) - f^\star\right] \leq \left(f(W^0) - f^\star\right)\left(1 - \tilde{\eta}\mu\left(1 - \lambda_{\max}\left[\mathbb{E}\left[I - H^t\right]\right] - 3L\tilde{\eta}\right)\right)^T$$

$$+ \frac{\tilde{\eta}L^2}{\tilde{\eta}\mu\left(1 - \lambda_{\max}\left[\mathbb{E}\left[I - H^t\right]\right] - 3L\tilde{\eta}\right)}\left(2\gamma^2 N^2 L\Delta^* + 2\gamma^2 NL\frac{1}{M}\sum_{m=1}^{M}\Delta_m^*\right)$$

$$+ \frac{L\tilde{\eta}^2}{\tilde{\eta}\mu\left(1 - \lambda_{\max}\left[\mathbb{E}\left[I - H^t\right]\right] - 3L\tilde{\eta}\right)}\frac{M-C}{C\max\{M-1,1\}}\left(2L\left(f\left(W^t\right) - f^\star\right) + 2L\Delta^*\right).$$

$\square$

