# OpenReview forum: "RAC-LoRA: A Theoretical Optimization Framework for Low-Rank Adaptation"
_ICLR.cc/2025/Conference — Submitted to ICLR 2025_

### Official Review · Reviewer_a6rr · 2024-10-29

**Soundness:** 2
**Presentation:** 3
**Contribution:** 2
**Rating:** 5
**Confidence:** 3

**Summary:**

This paper introduces RAC-LoRA, a new framework that addresses the convergence issues of Low-Rank Adaptation (LoRA) methods. It provides theoretical guarantees that RAC-LoRA can converge to the same solution as full-parameter fine-tuning (FPFT) but with reduced computational costs. The framework supports various optimization settings, including gradient descent and federated learning. Experimental results confirm the effectiveness of RAC-LoRA, showing competitive performance with other low-rank adaptation methods.

**Strengths:**

1. The paper introduces RAC-LoRA, a novel framework that rigorously addresses the convergence issues of existing Low-Rank Adaptation (LoRA) methods. It provides strong theoretical guarantees, including proofs of convergence rates for various optimization settings, which fills a critical gap in the literature.
2. RAC-LoRA is versatile and applicable to a wide range of machine learning tasks. The framework is tested on both traditional convex problems like logistic and linear regression and more complex tasks involving multilayer perceptrons (MLPs) on MNIST and RoBERTa on the GLUE benchmark.
3. The paper is exceptionally clear in its exposition, with important concepts and technical details explained thoroughly, making it accessible and valuable for both researchers and practitioners.

**Weaknesses:**

1. The paper primarily focuses on theoretical analysis and provides empirical validation to support its claims. However, the empirical comparisons with other state-of-the-art low-rank adaptation methods could be more extensive. Additional benchmarks and real-world datasets might further strengthen the practical relevance of the proposed framework.
2. Fixing one of the two low-rank matrices A and B in LoRA and training only the other can establish theoretical convergence guarantees, but it also partially diminishes the learning capability of LoRA.
3. Some typos: line 167: choose, line 342 $U=W^{t+1}$, $V=W^t$

**Questions:**

1. Fixing one of the two low-rank matrices A and B in LoRA and training only the other might impair the learning capability of LoRA. Could the authors provide more experimental results, such as using LoRA for SFT (Supervised Fine-Tuning) on GPT models? Would RAC-LoRA perform better in such tasks?

---

> ### Author Response · Authors · 2024-11-26
> **Authors' rebuttal (Part 1)**
>
> >Strengths:
>
> >1. The paper introduces RAC-LoRA, a novel framework that rigorously addresses the convergence issues of existing Low-Rank Adaptation (LoRA) methods. It provides strong theoretical guarantees, including proofs of convergence rates for various optimization settings, which fills a critical gap in the literature.
>
> >2. RAC-LoRA is versatile and applicable to a wide range of machine learning tasks. The framework is tested on both traditional convex problems like logistic and linear regression and more complex tasks involving multilayer perceptrons (MLPs) on MNIST and RoBERTa on the GLUE benchmark.
>
> >3. The paper is exceptionally clear in its exposition, with important concepts and technical details explained thoroughly, making it accessible and valuable for both researchers and practitioners.
>
> Thank you for your time and positive feedback!
>
> >Weaknesses:
>
> >1. The paper primarily focuses on theoretical analysis and provides empirical validation to support its claims. However, the empirical comparisons with other state-of-the-art low-rank adaptation methods could be more extensive. Additional benchmarks and real-world datasets might further strengthen the practical relevance of the proposed framework.
>
> Thank you for your question! Our primary motivation for this work was to provide a solid theoretical foundation for LoRA-type methods and to establish a rigorous understanding of their behavior. We focused on developing the theoretical framework and ensuring that our approach is grounded in robust principles.
>
> That said, we acknowledge that covering all aspects of such a broad topic in a single paper, especially within the constraints of a conference paper, is quite challenging. While our current work emphasizes theory, we agree that more extensive experimental results would further strengthen the contribution and provide practical validation.
>
> We are actively working on expanding our experimental evaluations, and this is currently in progress. We aim to explore additional setups and datasets to complement the theoretical findings and provide a more comprehensive perspective.
>
>
> >2. Fixing one of the two low-rank matrices A and B in LoRA and training only the other can establish theoretical convergence guarantees, but it also partially diminishes the learning capability of LoRA.
>
> Thank you for your question!
>
> To retain the learning capacity of the Asymmetric LoRA approach, as employed in RAC-LoRA, it is reasonable to double the rank. This adjustment compensates for the fact that one of the matrices in the decomposition is not learnable. By increasing the rank, the model effectively restores the expressive power that would otherwise be reduced due to the non-learnability of one matrix.
>
> This concept is thoroughly discussed in the paper by Zhu, Jiacheng, et al., *"Asymmetry in Low-Rank Adapters of Foundation Models"* (arXiv preprint arXiv:2402.16842, 2024). The authors analyze the asymmetry in low-rank adapters and show how adjusting the rank can address potential limitations while maintaining the method's overall effectiveness.
>
> >3. Some typos: line 167: choose, line 342 $U=W^{t+1}, V=W^t$
>
> Thank you for your feedback! We appreciate you pointing this out. We will fix the typos and add clear explanations to make the paper easier to understand. If you notice anything else that could be improved, please feel free to let us know. Thank you again for helping us improve our work!
>
> >Questions:
>
> >Fixing one of the two low-rank matrices A and B in LoRA and training only the other might impair the learning capability of LoRA. Could the authors provide more experimental results, such as using LoRA for SFT (Supervised Fine-Tuning) on GPT models? Would RAC-LoRA perform better in such tasks?
>
> Thank you for your thoughtful questions! Fixing one of the two low-rank matrices A or B in LoRA and training only the other can indeed impact the learning capability. As we mentioned earlier, to address this, doubling the rank in the Asymmetric LoRA approach used in RAC-LoRA is an effective solution. This adjustment compensates for the non-learnability of one matrix by increasing the overall capacity, allowing the model to recover the expressive power that would otherwise be lost.
>
> Regarding your suggestion for more experiments, such as using LoRA for supervised fine-tuning (SFT) on GPT models, we agree that this would be an interesting and valuable direction to explore. While we aim to include experiments on larger models, training GPT models requires substantial computational resources, which may limit the scope of such experiments in the current work. However, we plan to expand our experiments in this area as computational resources become available.
>
> Thank you once again for your suggestions—they are greatly appreciated and help us identify promising directions for future research!

---

> > ### Author Response · Authors · 2024-12-01
> > **Reply**
> >
> > Thank you for your time and effort in reviewing our work!
> >
> > We have made a concerted effort to prepare a rebuttal and clarify the issues you mentioned. We would greatly appreciate it if you could provide your feedback.
> >
> > If we have addressed the concerns you raised, would you kindly consider improving the score?

---

### Official Review · Reviewer_vvXw · 2024-10-30

**Soundness:** 2
**Presentation:** 3
**Contribution:** 2
**Rating:** 5
**Confidence:** 4

**Summary:**

This paper presents a novel parameter efficient fine-tuning dubbed RAC-LoRA, for fine-tuning the large models. Specifically, the authors theoretically analyze the smooth issue in the vanilla LoRA type of algorithms and give motivating example about how existing LoRA algorithm can fail easily in simple case. Additionally, the authors establish the convergence rates for the non-convex objectives with gradient descent, stochastic gradient descent and federated learning setting. To validate RAC-LoRA, the authors leverage simple to complex models with a few datasets. The results show that RAC-LoRA performs competitively with other low-rank adaptation methods.

**Strengths:**

1. The investigated topic is very interesting and critical. Particularly, fine-tuning large models is the most popular method to enable models performing downstream tasks.
2. This work is motivated well through examples and theoretical analysis, on top of the existing methods.
3. The literature survey looks extensive and thorough to include different LoRA variants.
4. The paper presents sufficient theoretical analysis to cover different objectives and algorithms for LoRA.
5. The proposed algorithm is validated through the empirical evidence.

**Weaknesses:**

1. The issue of Assumption 3.1. Though in this work, the authors point out the smoothness issue related to Assumption 3.1. However, they still use it, with Assumption 5.1. My first question is that, why the authors didn't directly address this issue to probably derive a new smoothness constant when the parameters have LoRA update.

2. The issue of Assumption 5.1. This assumption seems a bit problematic in this work. Since the smallest eigenvalue of a projection matrix is 0, as pointed out in the paper. Then, the authors mentioned that "the smallest eigenvalue of the expected value of the projection matrix can be strictly greater than zero". However, the expectation of a projection matrix is still a projection matrix, which would have the smallest eigenvalue being 0. Though in the Remark, the authors have verified this, I am still a bit confused about this.

3. Also, in Theorem E.2, the authors have defined the expected smoothness. Why do you need Assumption 3.1 again in the theorem statement? Typically, people would relax smoothness to expected smoothness, which is a weaker condition. Also, the notations in Assumption E.1. are not consistent.

4. The proof techniques are fairly standard, but with the help from Assumption 5.1. However, to me, such an assumption didn't really help resolve the issue of Loss of Lipschitz smoothness presented in the paper.

5. The experimental results on the image classification tasks to me are not promising. The authors should test their proposed algorithm on more complex datasets and larger models, such as CIFAR 100 with ResNet, instead of just MLP with MNIST. Also, it seems like freezing A matrix can yield the better results? From Table 2, the authors mentioned that "We find that RAC-LoRA performs competitively with other low-rank adaptation methods, but does not outperform Asymmetric LoRA despite having greater capacity". By looking at the results from Table, RAC-LoRA achieves 77.0 and Asymmetric LoRA gets 76.8. I wonder if the lower value, the better? Did I misunderstand here? It is good to see results on simple models and datasets, but to make the paper more technically solid and sound, more complex tasks are required.

6. Regarding the FL setting with RAC-LoRA, there are no results to validate the proposed algorithm and the corresponding analysis. The authors should present empirical evidence for the RAC-LoRA in the FL setting and show comparison with baselines.

**Questions:**

Please see questions in the above comments.

---

> ### Author Response · Authors · 2024-11-25
> **Authors' rebuttal (Part 1)**
>
> >Strengths:
>
> >1. The investigated topic is very interesting and critical. Particularly, fine-tuning large models is the most popular method to enable models performing downstream tasks.
>
> >2. This work is motivated well through examples and theoretical analysis, on top of the existing methods.
>
> >3. The literature survey looks extensive and thorough to include different LoRA variants.
>
> >4. The paper presents sufficient theoretical analysis to cover different objectives and algorithms for LoRA.
>
> >5. The proposed algorithm is validated through the empirical evidence.
>
> Thank you for your time and kind feedback!
>
> >Weaknesses:
>
> >1. The issue of Assumption 3.1. Though in this work, the authors point out the smoothness issue related to Assumption 3.1. However, they still use it, with Assumption 5.1. My first question is that, why the authors didn't directly address this issue to probably derive a new smoothness constant when the parameters have LoRA update.
>
> Thank you for your comment! There seems to be significant confusion regarding this point, so we would like to clarify it further.
>
> In the paper by Sun, Youbang, et al., *"Improving LoRA in Privacy-Preserving Federated Learning"* (arXiv preprint arXiv:2403.12313, 2024), it is rigorously proven that if the function $ f(W) $ is Lipschitz smooth, then the function $ f(W + BA) $ is **not** Lipschitz smooth for *any* constant. We emphasize once again that this result is **proven**, meaning it is theoretically impossible to derive any new or corrected constant for the LoRA update within this framework.
>
> In our work, we assume Lipschitz smoothness of the loss function, as it is a standard assumption in optimization theory and is widely used in related research. Recognizing this limitation, we propose a modification to the method that directly addresses this issue. Furthermore, we provide theoretical evidence showing that our approach resolves the problem and restores convergence.
>
> We hope this explanation helps to clarify the matter. We will include this clarification in the camera-ready version!
>
>
> >2. The issue of Assumption 5.1. This assumption seems a bit problematic in this work. Since the smallest eigenvalue of a projection matrix is 0, as pointed out in the paper. Then, the authors mentioned that "the smallest eigenvalue of the expected value of the projection matrix can be strictly greater than zero". However, the expectation of a projection matrix is still a projection matrix, which would have the smallest eigenvalue being 0. Though in the Remark, the authors have verified this, I am still a bit confused about this.
>
>
> Assume $r, d \in \mathbb{N}$ and $r<d$. For a uniformly chosen subspace $\mathbb{R}^r \subsetneq \mathbb{R}^d$ we define the orthogonal projection as $P: \mathbb{R}^d \mapsto \mathbb{R}^d$. Prove that $\mathrm{E}[P(\mathbf{v})] = \frac{r}{d} \mathbf{v}$
>
> **Proof:** We now aim to prove the correctness of our Remark.
>
> Consider any set of orthogonal bases $ \mathbf{e} = \left(\mathbf{e}\_1, \mathbf{e}\_2, \ldots, \mathbf{e}\_d\right) \in \mathbb{R}^{d \times d}$. From this set, we uniformly select a subset $\left(\mathbf{e}\_{d\_1}, \mathbf{e}\_{d\_2}, \ldots, \mathbf{e}\_{d\_r}\right)$ and define a subspace $\mathbb{R}^r$ spanned by these bases.
>
> The projected value of any vector $\mathbf{v}$ onto a basis vector $\mathbf{e}_j$ is given by $\mathbf{e}_j^{\prime} \mathbf{v}$. The corresponding vector component is then $\mathbf{e}_j \mathbf{e}_j^{\prime} \mathbf{v}$. Consequently, the orthogonal projection of $\mathbf{v}$ onto the subspace can be expressed as:
>
> $$
> P(\mathbf{v}) = \sum\_{j=1}^r \mathbf{e}\_{d\_j} \mathbf{e}\_{d\_j}^{\prime} \mathbf{v} = \mathbf{e D} \mathbf{e}^{\prime} \mathbf{v} \in \mathbb{R}^d
> $$
>
> Here, $\mathbf{D}$ is defined as a random diagonal matrix with $r$ ones and $(d-r)$ zeros on its diagonal. While the diagonal entries are not independent, their expectation is uniform, i.e., each entry has an expected value of $r / d$. Thus, the expectation of the projection is:
>
> $$
> \mathrm{E}[P(\mathbf{v})] = \mathrm{E}[\mathbf{e D} \mathbf{e}^{\prime}] \mathbf{v}.
> $$
>
> Using the **tower rule**, we proceed to simplify further. Since for any orthogonal matrix $\mathbf{e}$, it holds that $\mathbf{e}^{\prime} \mathbf{e} = \mathbf{I}$, we deduce:
>
> $$
> \left(\mathbf{e e}^{\prime}\right)^2 = \mathbf{e e}^{\prime} \mathbf{e e}^{\prime} = \mathbf{e}\left(\mathbf{e}^{\prime} \mathbf{e}\right) \mathbf{e}^{\prime} = \mathbf{e I e}^{\prime} = \mathbf{e e}^{\prime}.
> $$
>
> This implies that $\mathbf{e e}^{\prime} = \mathbf{I}$. Substituting this result back, we conclude:
>
> $$
> \mathrm{E}[P(\mathbf{v})] = \frac{r}{d} \mathrm{E}[\mathbf{e e}^{\prime}] \mathbf{v} = \frac{r}{d} \mathbf{v}.
> $$
>
> This result perfectly aligns with our initial claim.

---

> > ### Comment · Reviewer_vvXw · 2024-11-29
> > **Response to the rebuttal**
> >
> > Thanks for clarification from the authors and the technical proof here. I really appreciate that. The explanations here makes sense to me.

---

> ### Author Response · Authors · 2024-11-25
> **Authors' rebuttal (Part 2)**
>
> >2. The issue of Assumption 5.1. This assumption seems a bit problematic in this work. Since the smallest eigenvalue of a projection matrix is 0, as pointed out in the paper. Then, the authors mentioned that "the smallest eigenvalue of the expected value of the projection matrix can be strictly greater than zero". However, the expectation of a projection matrix is still a projection matrix, which would have the smallest eigenvalue being 0. Though in the Remark, the authors have verified this, I am still a bit confused about this.
>
> **Continue:**
>
> This proof demonstrates that the expected value of the projection matrix is the identity matrix, scaled by the factor $\frac{r}{d}$, where $r$ is the number of selected subspace dimensions and $d$ is the total dimensionality of the space. This result confirms that, in expectation, the projection spans the entire space, albeit scaled appropriately. Therefore, there is no contradiction in using this approach.
>
> The key insight here is that, while each individual projection focuses on a randomly selected subspace, the expectation of these projections effectively covers the entire space. This property is critical in ensuring that the optimization process using RAC-LoRA is unbiased with respect to the full parameter space. It is this ability to explore the whole space in expectation that allows RAC-LoRA to converge to the same solution as full parameter fine-tuning.
>
> In other words, RAC-LoRA’s random subspace selection scheme does not compromise the final outcome. Instead, it achieves computational efficiency by operating in smaller subspaces while maintaining theoretical guarantees for convergence to the optimal solution in the full parameter space. This proof solidifies the theoretical foundation for why RAC-LoRA works as intended.
>
> >3. Also, in Theorem E.2, the authors have defined the expected smoothness. Why do you need Assumption 3.1 again in the theorem statement? Typically, people would relax smoothness to expected smoothness, which is a weaker condition. Also, the notations in Assumption E.1. are not consistent.
>
> Thank you for your question! We understand that there may be some confusion regarding the expected smoothness assumption, and we are happy to clarify this point.
>
> The standard Lipschitz smoothness condition is an assumption about the *loss function*, specifically that its gradients are Lipschitz continuous. In contrast, the **expected smoothness** assumption concerns the behavior of the gradient estimator $g(W)$, and it can be seen as a generalization of the bounded variance assumption. These two assumptions serve different purposes but are both essential for rigorous analysis.
>
> In fact, in the standard analysis of SGD in the non-convex setting, both assumptions are commonly used. For reference, we point you to **Theorem 2** and **Theorem 3** in the paper by Khaled and Richtárik, *"Better Theory for SGD in the Nonconvex World"* (arXiv preprint arXiv:2002.03329, 2020). This paper discusses the necessity of these assumptions in detail. While the naming "expected smoothness" might initially seem confusing, it is widely accepted terminology in the field of SGD theory. For analyzing optimization methods in the non-convex setting, we require both the Lipschitz smoothness of the loss function and the expected smoothness of the gradient estimator. These assumptions together ensure convergence guarantees.
>
> We will add clarifications in the revised version of the paper to explicitly explain these assumptions, ensure consistent notation throughout the text, and provide further context to avoid any misunderstanding. Thank you for highlighting this, as it helps us improve the clarity of our work!

---

> > ### Comment · Reviewer_vvXw · 2024-11-29
> > **Response to rebuttal part 2**
> >
> > I also appreciate the clarification between the standard smoothness and expected smoothness, which is clear to me now.

---

> ### Author Response · Authors · 2024-11-25
> **Authors' rebuttal (Part 3)**
>
> >4. The proof techniques are fairly standard, but with the help from Assumption 5.1. However, to me, such an assumption didn't really help resolve the issue of Loss of Lipschitz smoothness presented in the paper.
>
> We respectfully but firmly disagree with this statement. It appears to reflect a personal opinion, and we find it somewhat unclear how best to address this concern. Our primary goal in this work was to establish and improve the convergence guarantees for LoRA-type methods. To achieve this, we provided a rigorous theoretical analysis, focusing on addressing key limitations. Specifically, we demonstrated that Assumption 5.1 is straightforward to satisfy under the uniform sampling of subspaces, as supported by our results.
>
> We are happy to provide further clarifications or additional details to ensure our approach and contributions are fully understood.
>
> >5. The experimental results on the image classification tasks to me are not promising. The authors should test their proposed algorithm on more complex datasets and larger models, such as CIFAR 100 with ResNet, instead of just MLP with MNIST. Also, it seems like freezing A matrix can yield the better results? From Table 2, the authors mentioned that "We find that RAC-LoRA performs competitively with other low-rank adaptation methods, but does not outperform Asymmetric LoRA despite having greater capacity". By looking at the results from Table, RAC-LoRA achieves 77.0 and Asymmetric LoRA gets 76.8. I wonder if the lower value, the better? Did I misunderstand here? It is good to see results on simple models and datasets, but to make the paper more technically solid and sound, more complex tasks are required.
>
> Thank you for your thoughtful question!
>
> In **Table 2**, we report the results based on the accuracy metric. While RAC-LoRA achieves an accuracy of 77.0 and Asymmetric LoRA achieves 76.8, we cannot conclusively state that one method outperforms the other, as the margin is too small to be statistically significant. In this context, higher accuracy is indeed better, but such a small difference falls within the range of potential experimental variability.
> We greatly appreciate your suggestion regarding additional experiments. We are planning to conduct further evaluations, including experiments on CIFAR-100 with ResNet, as you have proposed. This will provide a broader perspective on the comparative performance of the methods and help us ensure more robust conclusions.
>
> A thorough analysis of the impact of freezing matrix $A$ versus matrix $B$ is provided in the paper by Zhu, Jiacheng, et al., titled *"Asymmetry in Low-Rank Adapters of Foundation Models"* (arXiv preprint arXiv:2402.16842, 2024). This work delves into the asymmetry between these two approaches, highlighting their differences and exploring how freezing one matrix over the other affects the performance and behavior of LoRA-type methods.
>
> Thank you again for your feedback and valuable suggestions! If you have any other ideas or further recommendations, please feel free to share them.
>
>
> >6. Regarding the FL setting with RAC-LoRA, there are no results to validate the proposed algorithm and the corresponding analysis. The authors should present empirical evidence for the RAC-LoRA in the FL setting and show comparison with baselines.
>
> Thank you for your comment! You are correct that we do not currently include experiments in the Federated Learning (FL) setting. This is an important area, but FL is quite complex and comes with unique challenges, such as client heterogeneity, communication efficiency, privacy concerns, and scalability. Properly addressing these aspects would require a separate, dedicated paper.
>
> Our current work is already dense, and adding a full FL study would go beyond the scope of this paper. However, we agree that FL experiments are an exciting direction for future work, and we plan to explore this in more detail later.
>
> That said, we could add some preliminary FL experiments in the camera-ready version to provide a basic idea of how our method might perform in such a setting. This would give readers a starting point while keeping the main focus of the paper clear.

---

> > ### Comment · Reviewer_vvXw · 2024-11-29
> > **Response to rebuttal part 3**
> >
> > After carefully reading the rebuttal from the authors, I think they have addressed some of my concerns. I really appreciate their efforts and time to put the detailed rebuttal together. But taking other reviewers' comments into consideration, I will still maintain the current score. Since the authors did not upload a revised draft to show the changes. Also, the empirical concerns still remain without any new results.

---

> > > ### Author Response · Authors · 2024-12-01
> > > **Reply**
> > >
> > > Thank you for your response!
> > >
> > > Please note that the discussion period and the time allocated for updating the manuscript were quite short. Given this, we decided to prioritize preparing a rebuttal to clarify the issues rather than making manuscript updates, as doing so within such a brief timeframe would not have been effective. As you mentioned, we have successfully addressed the questions raised. We apologize for not having enough time to incorporate these updates into the manuscript, but we plan to do so for the camera-ready version.
> > >
> > > Additionally, we have also clarified the questions raised by other reviewers.
> > >
> > > Given these points, could you kindly consider improving the score? Also, we would appreciate it if you could provide guidance on what further actions we can take to increase the score from our side.

---

### Official Review · Reviewer_k2xB · 2024-10-31

**Soundness:** 3
**Presentation:** 3
**Contribution:** 2
**Rating:** 5
**Confidence:** 2

**Summary:**

This paper proposes a new Low-Rank Adaptation algorithm RAC-LORA and theoretically proves its convergence, which is lacking in other Low-Rank Adaptation algorithms. Experiments validate the effectiveness of the proposed algorithm.

**Strengths:**

1. The authors provide rigorous proofs for the convergence and convergence rate of RAC-LORA under the smoothness and PL conditions.
2. The authors demonstrate the superiority of their method for convex problems and problems under the condition of restricting the capacity of the low-rank adaptations.

**Weaknesses:**

For NLP tasks, the algorithm proposed by the authors shows no significant advantage in performance compared to existing methods. The authors did not discuss other potential advantages of the algorithm in depth, aside from the convergence guarantee of their algorithm. Please see the below comments about other weaknesses.

**Questions:**

Significant issues
1. Could the authors clarify how the final output of RAC-LORA is selected in practice? Is it the low-rank matrix chosen uniformly at random according to the theory, or from the last iteration? What is the difference in performance between these two approaches in practice?
2. The process of RAC-LORA seems like unfolding the gradient-based optimization algorithm for solving Problem (1) in Line 104, where each part of the chain can be understood as a single step of gradient descent. Could the authors further explain the relationship between each step of RAC-LORA and each step of gradient descent used in LoRA, as well as the role of using a new randomized matrix at each step?
3. RAC-LORA requires calculating the pseudoinverse each step. Could the authors comment on its time efficiency compared to other algorithms?

Minor issues
1. Could the authors provide the performance of other algorithms on convex problems as discussed in Section 6.1? Is their performance similar to that in Figure 1?
2. In Line 342, it seems that letting $U=W^t, V=B_S^t \hat{A}^t$ does not lead to the inequality in Line 343.
3. Could the authors explain the meaning of "iteration" in Figure 4?

---

> ### Author Response · Authors · 2024-11-25
> **Authors' rebuttal (Part 1)**
>
> >Strengths:
>
> >The authors provide rigorous proofs for the convergence and convergence rate of RAC-LORA under the smoothness and PL conditions.
>
> >The authors demonstrate the superiority of their method for convex problems and problems under the condition of restricting the capacity of the low-rank adaptations.
>
> Thank you for your time and positive feedback!
>
> >Weaknesses:
>
> >For NLP tasks, the algorithm proposed by the authors shows no significant advantage in performance compared to existing methods. The authors did not discuss other potential advantages of the algorithm in depth, aside from the convergence guarantee of their algorithm. Please see the below comments about other weaknesses.
>
> Our primary motivation was to develop a theoretically grounded method. Through our analysis, we demonstrated that the standard LoRA method encounters significant convergence issues, even in simple settings such as small-dimensional quadratic problems. To address this, we introduced our new method, which resolves these issues effectively.
>
> For NLP tasks, we observed that our method achieves results comparable to those of classical LoRA. However, unlike the heuristic nature of the classical approach, our method is backed by strong theoretical guarantees. This means we successfully achieved our objective: to design a theoretically sound method that maintains competitive performance in practical applications without compromising on efficiency or results.
>
> >Questions:
>
> >Significant issues
>
> >1. Could the authors clarify how the final output of RAC-LORA is selected in practice? Is it the low-rank matrix chosen uniformly at random according to the theory, or from the last iteration? What is the difference in performance between these two approaches in practice?
>
> Thank you for your question!
>
> Please note that in the case of the PL condition, we provide guarantees for the last iterate, and we have conducted experiments in a controlled setting to validate this. These results are shown in Figures 2 and 3.
>
> For the general non-convex setting, it is true that our theoretical guarantee applies to a model chosen uniformly at random from all iterations. This is due to the fact that the general Lipschitz-smooth non-convex case represents an extremely broad class of functions, making it challenging to guarantee convergence for the last iterate in this setup.
>
> In practice, it is not feasible to store all models in memory to randomly choose from them, so we keep the last one. However, it is an interesting idea to develop a procedure to select the best model across all iterations, but we leave this for future work.
>
> >2. The process of RAC-LORA seems like unfolding the gradient-based optimization algorithm for solving Problem (1) in Line 104, where each part of the chain can be understood as a single step of gradient descent. Could the authors further explain the relationship between each step of RAC-LORA and each step of gradient descent used in LoRA, as well as the role of using a new randomized matrix at each step?
>
> Thank you for the question! Let's explain this relationship in detail.
> The LoRA (Low-Rank Adaptation) approach works by restricting the optimization process to a specific, low-dimensional subspace. In other words, LoRA reduces the number of parameters being optimized compared to the full model, thereby finding a solution within this smaller subspace. Each gradient descent step, in this case, is directed toward a solution within this predefined, fixed subspace.
> In contrast, the RAC-LoRA method introduces randomness into this process. Specifically, for each gradient step, it samples a random sketch and uses it to compute a random projection matrix. This matrix determines a new subspace for optimization, meaning that each step occurs in a dynamically and randomly chosen subspace.
> The key role of the random projection in RAC-LoRA is that it allows the optimization process to explore multiple subspaces rather than being confined to a single one, as in standard LoRA. By sampling and optimizing in diverse subspaces, RAC-LoRA mimics the effect of training in the full parameter space, potentially leading to a solution that is equivalent to full parameter training. This approach helps mitigate the risk of being trapped in a suboptimal region defined by a fixed subspace.

---

> > ### Comment · Reviewer_k2xB · 2024-11-28
> > **Response to Rebuttal**
> >
> > I appreciate the authors taking the time to respond to my questions. In my view, an important distinction between RAC-LoRA and existing algorithms is that the authors add a random projection step in each gradient descent step. Therefore, I think it would be good to discuss the practical implementation of the random projection at line 362, such as the computational complexity, the runtime, and the approximate projection mentioned by the authors.

---

> > > ### Author Response · Authors · 2024-12-01
> > > **Reply**
> > >
> > > Thank you for you response!
> > >
> > > Thank you for suggesting a comparison of computational time, projection time, and other relevant aspects. While we appreciate this valuable idea, we believe that measuring real-time performance accurately can be challenging, as the actual working time may vary considerably depending on implementation details and hardware characteristics. In our experiments, we chose to use epochs for comparison, as the time per epoch was roughly consistent across different runs, providing a more stable metric.
> > >
> > > That said, we fully recognize the importance of assessing real working time and other relevant performance metrics. We will certainly consider incorporating these metrics in future analyses to offer a more comprehensive evaluation.
> > >
> > > Additionally, we plan to conduct a thorough investigation into the projection time aspects. Once again, thank you for your insightful suggestion, and we will explore ways to improve this aspect in our work.
> > >
> > > If we have successfully addressed the issues raised in your response, we kindly ask that you consider increasing the score.

---

> ### Author Response · Authors · 2024-11-25
> **Authors' rebuttal (Part 2)**
>
> >Significant issues
>
> >3. RAC-LORA requires calculating the pseudoinverse each step. Could the authors comment on its time efficiency compared to other algorithms?
>
> Thank you for this question! Let’s clarify this aspect further.
>
> In this context, it’s important to note that we do not need to explicitly compute the pseudo-inverse itself. Instead, our goal is to obtain the projection matrix, which can often be implemented more efficiently by leveraging its specific structure. This avoids the computational burden associated with directly calculating the pseudo-inverse, especially in high-dimensional settings.
>
> Additionally, it’s worth emphasizing that we don’t necessarily require the exact projection matrix HtHt. Instead, we can use an approximate version of this matrix. As long as the conditions outlined in Assumption 5.1 hold, the method is guaranteed to converge. This flexibility allows for further optimization and simplification of the implementation, potentially reducing computational overhead while maintaining theoretical guarantees.
>
> We will provide additional clarifications and details on this aspect, including how approximations to $H^t$ can be constructed and the practical trade-offs involved. By adopting approximate matrices, the method becomes more scalable, especially for large-scale problems, without compromising its convergence properties.
>
> >minor issues
>
> >1. Could the authors provide the performance of other algorithms on convex problems as discussed in Section 6.1? Is their performance similar to that in Figure 1?
>
> Thank you for the suggestion! We appreciate your feedback and will incorporate such plots in the camera-ready version of the paper.
> We have already explored this approach during our experiments, and the observed behavior was consistent with what is shown in Figure 1. Specifically, it demonstrated a lack of convergence, reinforcing the conclusions we drew from our current analysis.
>
> >2. $\text { In Line } 342 \text {, it seems that letting } U=W^t, V=B_S^t \hat{A}^t \text { does not lead to the inequality in Line } 343 .$
>
> Yes, there is indeed a typo in line 342. Thank you for bringing this to our attention! We will correct it in the next revision to ensure clarity and accuracy.
>
> >3. Could the authors explain the meaning of "iteration" in Figure 4?
>
> Thank you for pointing this out! In **Figure 4**, *iteration* refers to the number of **LoRA blocks** in the chain. Each iteration corresponds to adding another LoRA block, showing how performance evolves as more blocks are included.
> We will add this clarification to the text or figure caption in the next version to ensure clarity. Thank you for your helpful feedback!

---

### Official Review · Reviewer_jQG8 · 2024-11-05

**Soundness:** 2
**Presentation:** 2
**Contribution:** 3
**Rating:** 5
**Confidence:** 3

**Summary:**

This paper deals with the convergence issue in LoRA, by proposing a method that randomly initializes one matrice (A) and fixes it, while fully training the other one (B), in every step. The authors provide both empirical and theoretical motivations for this convergence issue of LoRA, and provide theoretical convergence guarantees for the proposed method. The proposed optimization method is evaluated on both convex problems, like simulated linear regression and classification problems, and also non-convex finetuning of language models.

**Strengths:**

1. This paper’s motivation is well-justified and tries to solve an important problem in the LoRA domain.
2. The solution is relatively simple and easy to implement.

**Weaknesses:**

1. There is still a gap between the theoretical analysis and the real LoRA setting, and the current analysis only works on smooth problems.

2. There is no evaluation in terms of training efficiency (beyond the number of parameters). Given now A or B needs to be fully trained, I assume training will be more time-cosuming. It would be nice to show the training curve and the convergence pattern in the NLP task as well. It would be nice to discuss what is the potential "cost" of utilizing this method, comparing to the original LoRA.

3. Proof of the analysis may depend on previous work, but it is not clear in the main paper, if we do not try to tease it out from the appendix. It would be nice to state clearly how things are proved so that it is easier to evaluate any theoretical contribution.

4.  The presentation in the technical sections is not very consistent and sound. See questions.

**Questions:**

1. In line 368, you mentioned the “smallest eigenvalues’’ in 5.1 but I believe there is no mention of that in 5.1. There is also a typo in line 352.

2. When discussing the related work in LoRA, the authors mentiond that “LoRA is highly sensitive to the choice of the hyper-parameters. A good theory should be able to explain or remove this issue.” So, for the proposed method, I was wondering whether the theory provides a recommended stepsize? How does it work in practice?

---

> ### Author Response · Authors · 2024-11-25
> **Authors' rebuttal (Part 1)**
>
> >Strengths:
>
> >This paper’s motivation is well-justified and tries to solve an important problem in the LoRA domain.
>
> >The solution is relatively simple and easy to implement.
>
> Thank you for your time and effort spent reviewing and positive comments!
>
> >Weaknesses:
>
> >1. There is still a gap between the theoretical analysis and the real LoRA setting, and the current analysis only works on smooth problems.
>
> We respectfully but firmly disagree with this statement. The smoothness assumption is a fundamental and widely accepted concept in optimization theory. It has been extensively used to analyze gradient-based methods, as evidenced in numerous foundational works such as the General SGD paper, asynchronous SGD, and sparsity-based SGD (e.g., Shi, Shaohuai, et al. *"A convergence analysis of distributed SGD with communication-efficient gradient sparsification."* IJCAI, 2019), as well as analyses of Local and Minibatch SGD (e.g., Woodworth, Blake, et al. *"Is local SGD better than minibatch SGD?"* ICML, 2020). The literature is vast, and there are countless examples where the Lipschitz smoothness assumption has been instrumental in advancing the field of continuous optimization.
>
> Criticizing the Lipschitz smoothness assumption amounts to questioning a cornerstone of the entire field of continuous optimization. Moreover, this assumption is explicitly utilized in the analysis of methods like LoRA, as demonstrated in Sun, Youbang, et al. *"Improving LoRA in privacy-preserving federated learning."* (arXiv:2403.12313, 2024).
>
> We strongly believe that the Lipschitz smoothness assumption is an appropriate starting point for theoretical analysis, particularly in foundational work on low-rank adaptation. While we recognize the value of more generalized smoothness assumptions, such as those discussed in Li, Haochuan, et al. *"Convex and non-convex optimization under generalized smoothness."* (NeurIPS 2024), we argue that beginning with the classical and well-established L-smoothness framework provides clarity and a solid baseline. Extensions to generalized smoothness assumptions can be an excellent direction for future research.
>
>
> >2. There is no evaluation in terms of training efficiency (beyond the number of parameters). Given now A or B needs to be fully trained, I assume training will be more time-cosuming. It would be nice to show the training curve and the convergence pattern in the NLP task as well. It would be nice to discuss what is the potential "cost" of utilizing this method, comparing to the original LoRA.
>
>
> Thank you for your thoughtful question! Let us provide a more detailed clarification.
>
> To ensure a fair comparison, we set the **same number of epochs** (i.e., passes over the dataset) for both the proposed method and the classical LoRA. By fixing the number of epochs, we can consistently evaluate and control the computational efficiency of both approaches. This ensures that any observed differences in performance are not due to variations in training duration.
>
> Regarding the statement:
> > "Given now A or B needs to be fully trained, I assume training will be more time-consuming."
>
> We would like to clarify that this assumption is not accurate. In our algorithm, **A or B does not need to be fully trained**. As outlined in Algorithm 1 (line 4), we only require an *approximate solution* to the inner minimization problem. Importantly, this approximate solution can be obtained with just a very small number of epochs—or, in some cases, even fewer optimization steps. This design ensures that the method remains computationally efficient. Furthermore, we explicitly set the same number of epochs for our method as for the classical LoRA method, ensuring there are no additional training costs associated with our approach.
>
> To further support this, we have included convergence plots for NLP tasks in the appendix. We encourage you to refer to **Figure 4** for a more detailed comparison of the convergence behavior across methods.
>
> Lastly, we appreciate your suggestion to discuss the “cost” of the approach. However, it is not entirely clear what specific metric you are referring to when mentioning “cost.” Could you please clarify whether you mean computational cost, memory usage, or another metric? Specifying this would help us address your concern more directly. For now, we emphasize that by maintaining the same number of epochs across methods, we ensure a fair and unbiased comparison while avoiding any additional computational costs.
>
> We hope this explanation provides clarity, and we’re happy to address further questions or concerns!

---

> > ### Comment · Reviewer_jQG8 · 2024-11-27
> > **Response**
> >
> > I thank the authors for their rebuttal. I have several responses.
> >
> > 1. Even though a lot of work and maybe most work in studying optimization in this setting starts with smooth assumptions, my statement that "There is still a gap between the theoretical analysis and the real LoRA setting" is still true. I can accept that the authors follow previous work and start with such assumptions. And it also can be true that this gap is hard to merge now. But I feel admitting the gap is reasonable. This can be a too harsh criticism (maybe a criticism to many previous works that I have).
> >
> > 2. Related to the previous point, in the paper, line 300, the text reads "fully trained". I can understand the point that, maybe A and B does not need to be fully trained in practice. But potentially the theory needs it. That is also a "gap" between the theory and practice in my opinion.
> >
> > 3. By cost, I mean "training time", as mentioned in my comment. For example, for training the model to achieve the same performance, does it take longer than previous methods. I think we cannot just compare number of epochs, as in this case, each epoch may take different time.
> >
> > In general, I liked the work in the sense it tries to use theory to motivate the design of algorithms. I just want to point out that it is good to talk about limitations.

---

> > > ### Author Response · Authors · 2024-12-01
> > > **Reply**
> > >
> > > Thank you for your thoughtful response. We appreciate your feedback and would like to address the points you raised.
> > >
> > > **1.** The gap between optimization theory and deep learning practice indeed exists. We will include a clarification remark on this in the revised manuscript.
> > > At first glance, the Lipschitz smoothness assumption might seem restrictive and unnecessary. However, as we elaborate in the paper, even under this foundational assumption, existing algorithmic developments remain limited and leave significant gaps that need to be addressed.
> > >
> > > We believe that if a method encounters issues even in a simplified setting, these should be resolved first. Otherwise, the method risks remaining a heuristic approach without a strong theoretical foundation.
> > >
> > > Additionally, the well-known optimization method Adam, which incorporates momentum, was inspired by the works of Nesterov and Polyak. Theoretically, momentum methods improve complexity mainly in the convex setting—a much more restrictive assumption in the context of deep learning. In non-convex cases, momentum does not offer benefits in terms of theoretical complexity guarantees. This suggests that simplified settings, such as those with a convexity assumption, often better model real-world applications than overly generalized approaches.
> > >
> > > Polyak, B. T. (1964). *Some methods of speeding up the convergence of iteration methods*. USSR Computational Mathematics and Mathematical Physics, 4(5), 1-17.
> > >
> > > Yurii Nesterov, "A Method for Solving the Convex Programming Problem with Convergence Rate \(O(1/k^2)\)," *Soviet Mathematics Doklady*, vol. 27, pp. 372–376, 1983.
> > >
> > > In general, we believe that the analysis of methods should begin with standard, well-established settings, which have demonstrated their utility, before moving on to more specific or complex cases.
> > >
> > > We also thank you for suggesting the exploration of more general smoothness assumptions. We will consider this topic in future work.
> > >
> > > **2.** Apologies for the confusion. What we intended to convey is that while one matrix is randomized, the other matrix is trainable without any constraints. We will ensure this is clarified in line 300 to avoid further misunderstanding.
> > >
> > > It is also important to highlight that our analysis does not require fully training each matrix in the chain (which could be quite lengthy). Instead, it is sufficient to train these matrices for only a few epochs or even a small number of iterations. This approach significantly reduces computational overhead while still maintaining the validity of our theoretical analysis.
> > >
> > > Please note once again that we do not need to fully train the matrix, and our analysis does not require this anywhere.
> > >
> > > We encourage you to review Section D, particularly Corollaries D.2.1 and D.3.1, where we provide a detailed analysis of cases where the matrices in the chain are trained using only a few gradient-type steps. This demonstrates that our approach is efficient and does not demand fully optimized matrices.
> > >
> > > Additionally, as stated explicitly in line 4 of Algorithm 1, the optimization subproblem for training the matrix in the chain is designed to be approximate. This further underscores that fully training the matrix is unnecessary, as our analysis accommodates such approximations seamlessly.
> > >
> > > We hope this clarification resolves any remaining concerns, and we are happy to address any further questions.
> > >
> > > **3.** Thank you for suggesting a comparison of the time. While we appreciate the idea, we believe it is challenging to measure real-time performance accurately, as the actual working time can vary significantly depending on implementation details and hardware aspects. In our experiments, we opted to use epochs for comparison, as the time per epoch was approximately the same across different runs, making it a more consistent metric.
> > >
> > > However, we fully recognize the value of measuring real working time or other more applicable performance metrics. We will certainly consider incorporating such metrics in future analyses to provide a more comprehensive evaluation. Thank you again for your insightful suggestion, and we will explore ways to improve this aspect in our work.
> > >
> > > If we address the issues you raised, we kindly ask you to consider increasing the score.

---

> ### Author Response · Authors · 2024-11-25
> **Authors' rebuttal (Part 2)**
>
> >Weaknesses:
>
> >3. Proof of the analysis may depend on previous work, but it is not clear in the main paper, if we do not try to tease it out from the appendix. It would be nice to state clearly how things are proved so that it is easier to evaluate any theoretical contribution.
>
> Please note that a portion of the convergence proof is presented in *Section 5.1*. However, due to the page limit constraints, we were unable to include the full proof in the main body of the paper. While we aimed to prioritize clarity and conciseness in presenting the key results, we understand that a more detailed explanation would be beneficial for readers.
> That said, we truly value your feedback. Based on your suggestion, we will explore the possibility of rearranging some parts of the paper to make room for more proof details in the main text. We believe this adjustment could enhance the clarity and accessibility of the theoretical contributions for the audience.
>
> >4. The presentation in the technical sections is not very consistent and sound. See questions.
>
> Thank you for your comment! We truly appreciate your attention to detail and your feedback. We will carefully review the paper to identify and address any typos, inconsistencies, or other issues to ensure the manuscript is polished and consistent throughout.
>
> >Questions:
>
> >1. In line 368, you mentioned the “smallest eigenvalues’’ in 5.1 but I believe there is no mention of that in 5.1. There is also a typo in line 352.
>
> Thank you for your comment! Let us clarify this aspect in more detail.
> We indeed explicitly mentioned the smallest eigenvalues in the context of our analysis. This is formally addressed and rigorously defined in Assumption 5.1, where we provide a clear and precise formulation. This assumption plays a critical role in our theoretical framework and serves as the foundation for the convergence analysis presented in the paper.
>
> Assumption 5.1. Consider a projection matrix H generated by Left Sketch 4.1 or Right Sketch 4.2. Assume that the sampling distributions $\mathcal{D}_S^B$ and $\mathcal{D}_S^A$ are such that the smallest eigenvalue of the expected projection matrix $H$ generated by sampled matrix is positive:
>
> $$
> \lambda_{\min }^H=\lambda_{\min }[\mathbb{E}[H]]>0 .
> $$
>
>
> To ensure greater clarity, we will add additional explanations and elaborations in Section 5.1. This will include a more detailed discussion on how the smallest eigenvalues influence the behavior of the algorithm and its convergence properties, as well as any relevant insights to strengthen the reader’s understanding.
>
> >2. When discussing the related work in LoRA, the authors mentiond that “LoRA is highly sensitive to the choice of the hyper-parameters. A good theory should be able to explain or remove this issue.” So, for the proposed method, I was wondering whether the theory provides a recommended stepsize? How does it work in practice?
>
>
> Thank you for the question! Let us provide a detailed clarification.
>
> The recommended step size provided by the theoretical analysis is $ \frac{1}{L} $, where $ L $ is the Lipschitz constant of the gradient. This step size is a widely recognized and standard choice for both Gradient Descent (GD) and Stochastic Gradient Descent (SGD) methods in the optimization literature. Its simplicity and theoretical grounding make it a natural choice for our method as well.
>
> An important advantage of our approach lies in its simplicity regarding parameter tuning. Unlike the standard LoRA approach, which requires tuning **two parameters**—the learning rate and the scaling parameter $ \alpha $—our method requires tuning only a **single parameter**, the step size. This significantly reduces the sensitivity of the method to hyperparameter choices and makes it easier to use in practice. Fewer parameters mean fewer hyperparameter search trials, which is especially beneficial in resource-constrained settings or when training on large-scale datasets.
>
> Additionally, the reduced dependence on multiple parameters enhances the robustness of our method. In many real-world scenarios, practitioners often face challenges in fine-tuning multiple hyperparameters, which can impact the reproducibility and efficiency of the training process. By requiring only one parameter to tune, our approach simplifies this process, making it more user-friendly and accessible.
>
> We believe this practical advantage, combined with the strong theoretical backing, highlights the usability and efficiency of our method. If you have further questions or suggestions, we would be happy to elaborate further!

---

### Meta-Review · Area_Chair_f2nt · 2024-12-23

**Metareview:**

This paper attempts to "shine some light on LoRA's convergence issues" (section 3). It analyzes the update process of LoRA matrices during Gradient descent highlighting loss of Lipszhitz continuity of the gradient.  It then introduces a new assymetric LoRa mechanism to enhance convergence while preserving some of LoRAs attractive features: modeling flexibility and efficiency.

Strengths: strong theoretical guarantees, including proofs of convergence rates for various optimization settings.

Weaknesses: empirical comparisons with other state-of-the-art low-rank adaptation methods could be more extensive; evaluation of training efficiency; likely due to technical presentation issues, the proof and validity of assumptions made was not clear to multiple reviewers.

**Additional Comments On Reviewer Discussion:**

This paper attempts to bring some optimization analysis rigor into LoRA literature which is a valuable contribution. Unfortunately though, none of the reviewers moved away from leaning to reject during the discussion phase. Some discussion focused on whether LoRA analysis under common assumptions made in optimization literature such as smoothness would be of practical value. Another concern raised was:  the algorithm proposed by the authors shows no significant advantage in performance compared to existing methods on NLP tasks. The fact that this approach is backed by theory but the baseline is not was not convincing enough. The experimental results on the image classification tasks were also not viewed as promising; the authors acknowledge that "we cannot conclusively state that one method outperforms the other, as the margin is too small to be statistically significant". The weaknesses consistently raised on empirical results weighed significantly in this decision.

---

### Decision · Program_Chairs · 2025-01-22

Reject